# Understanding and improving model representation of aerosol optical properties for a Chinese haze event measured during KORUS-AQ

Pablo E. Saide[1, 2], Meng Gao[3], Zifeng Lu[4], Daniel L. Goldberg[4], David G. Streets[4], Jung-Hun Woo[5], Andreas Beyersdorf[6], Chelsea A. Corr[7], Kenneth L. Thornhill[8], Bruce Anderson[8], Johnathan W. Hair[8], Amin R. Nehrir[8], Glenn S. Diskin[8], Jose L. Jimenez[9], Benjamin A. Nault[9], Pedro Campuzano-Jost[9], Jack Dibb[10], Eric Heim[10], Kara D. Lamb[11], Joshua P. Schwarz[11], Anne E. Perring[12], Jhoon Kim[13], Myungje Choi[13, 14], Brent Holben[15], Gabriele Pfister[16], Alma Hodzic[16], Gregory R. Carmichael[17], Louisa Emmons[16], and James H. Crawford[8]

[1]Department of Atmospheric and Oceanic Sciences, University of California – Los Angeles, Los Angeles, CA, USA
[2]Institute of the Environment and Sustainability, University of California – Los Angeles, Los Angeles, CA, USA
[3]Department of Geography, Hong Kong Baptist University, Hong Kong SAR, China
[4]Energy Systems Division, Argonne National Laboratory, Argonne, IL 60439, USA
[5]Department of Technology Fusion Engineering, Konkuk University, Seoul, South Korea
[6]Department of Chemistry & Biochemistry, California State University San Bernardino, San Bernardino, CA, USA.
[7]USDA UV-B Monitoring and Research Program, Natural Resource Ecology Laboratory, Colorado State University, Fort Collins, CO, USA.
[8]NASA Langley Research Center, Hampton, VA, USA
[9]Department of Chemistry, and Cooperative Institute for Research in Environmental Sciences, University of Colorado, Boulder, CO, USA
[10]Institute for the Study of Earth, Oceans, and Space, University of New Hampshire, Durham, NH, USA
[11]Earth System Research Laboratory, NOAA, Boulder, CO, USA
[12]Department of Chemistry, Colgate University, Hamilton, NY, USA
[13]Department of Atmospheric Sciences, Yonsei University, Seoul, 03722, Korea
[14]NASA Jet Propulsion Laboratory, Pasadena, California, USA.
[15]NASA Goddard Space Flight Center, Greenbelt, Maryland, USA.
[16]Atmospheric Chemistry Observations and Modeling Lab, National Center for Atmospheric Research, Boulder, CO, USA
[17]Center for Global & Regional Environmental Research, University of Iowa, Iowa City, Iowa, USA

*Correspondence to*: Pablo E. Saide (saide@atmos.ucla.edu)

**Abstract.** KORUS-AQ was an international cooperative air quality field study in South Korea that measured local and remote sources of air pollution affecting the Korean peninsula during May-June 2016. Some of the largest aerosol mass concentrations were measured during a Chinese haze transport event (May 24[th]). Air quality forecasts using the WRF-Chem model with aerosol optical depth (AOD) data assimilation captured AOD during this pollution episode but over-predicted surface particulate matter concentrations in South Korea, especially $PM_{2.5}$ often by a factor of 2 or larger. Analysis revealed multiple sources of model deficiency related to the calculation of optical properties from aerosol mass that explain these discrepancies. Using in-situ observations of aerosol size and composition as inputs to the optical properties calculations showed that using a low resolution size bin representation (4 bins) under-estimates the efficiency at which aerosols scatter

and absorb light (mass extinction efficiency). Besides using finer-resolution size bins (8-16 bins), it was also necessary to increase the refractive indices and hygroscopicity of select aerosol species within the range of values reported in the literature to achieve better consistency with measured values of mass extinction efficiency (6.7 $m^2$/g observed average) and light scattering enhancement factor (f(RH)) due to aerosol hygroscopic growth (2.2 observed average). Furthermore, evaluation of optical properties obtained using modeled aerosol properties revealed the inability of sectional and modal aerosol representations in WRF-Chem to properly reproduce the observed size distribution, with the models displaying a much wider accumulation mode. Other model deficiencies included an under-estimate of organic aerosol density (1.0 $g/cm^3$ in the model vs. and observed average of 1.5 $g/cm^3$) and an over-prediction of the fractional contribution of submicron inorganic aerosols other than sulfate, ammonium, nitrate, chloride and sodium corresponding to mostly dust (17-28% modeled vs. 12% estimated from observations). These results illustrate the complexity of achieving an accurate model representation of optical properties and provide potential solutions that are relevant to multiple disciplines and applications such as air quality forecasts, health impact assessments, climate projections, solar-power forecasts, and aerosol data assimilation.

## 1. Introduction

Exposure to air pollutants is estimated to be the leading environmental risk affecting human health (Gakidou et al., 2017) and aerosols represent the leading pollutant responsible for these effects (Cohen et al., 2017). The estimates of aerosol impacts on human health generally involve the use of ground-based monitoring networks that are combined with aerosol optical depth (AOD) satellite retrievals and/or model estimates for regions not monitored to obtain global estimates (Liu et al., 2009; van Donkelaar et al., 2016; Goldberg et al., 2019a). The use of AOD to estimate surface concentrations often involves using atmospheric composition simulations able to "translate" a column-integrated measure of light extinction due to aerosols (AOD) into surface aerosol mass concentrations. Satellite AOD is also often used to improve air quality forecasts of surface particulate matter through data assimilation (Saide et al., 2013; Saide et al., 2014; Kumar et al., 2019; Benedetti et al., 2009). Aerosols also impact climate through aerosol-cloud-radiation interactions and represent one of the largest uncertainties in climate projections (Boucher et al., 2013). Chemistry-climate models estimate 3-D distributions of aerosols, which are used by the radiative transfer module to estimate aerosol radiative effects. Again, this translation of aerosol mass to optical properties is performed in these models, often showing large inter-model variability (Myhre et al., 2013; Stier et al., 2013; Kipling et al., 2016). Similarly, short term predictions of solar power (Schroedter-Homscheidt et al., 2013; Jimenez et al., 2016) and visibility forecasts (Clark et al., 2008; Lee et al., 2016a) also require the use of aerosol optical properties. Thus, evaluating the ability of models to properly translate aerosol mass and number concentrations into aerosol optical properties is key to providing confidence in model results supporting these disciplines.

Previous research has shown various degrees of consistency between model evaluations of surface aerosol mass concentrations and column integrated aerosol properties. Lee et al. (2016b) used satellite AOD to constrain surface $PM_{10}$

(particulate matter with diameters below 10 μm) predictions and found large improvements against surface monitors regardless of the aerosol optical properties model used. Their results also showed slight discrepancies in the $PM_{10}$ and AOD when comparing models to observations, with some optical models showing larger biases in $PM_{10}$ than AOD and some presenting the opposite behavior. Lennartson et al. (2018) also found discrepancies when comparing the ratio between $PM_{2.5}$ and AOD for observations and WRF-Chem simulations, finding the modeled ratios were 30-50% higher over South Korea

during May-June 2016. Similar discrepancies were found by Mangold et al. (2011) when assessing model skill in predicting a regional pollution event over Europe driven by forest fire emissions and stagnation. Crippa et al. (2019) performed an ensemble of simulations to assess what combination of model inputs and configurations resulted in the best agreement to observations in the southeast US. They reported that simulations configured with a modal aerosol model performed the best against AOD observations, while a sectional aerosol approach showed the best agreement against surface PM2.5,

hypothesizing that aerosol hygroscopic growth and optical properties calculations could play a role in this discrepancy. Palacios-Peña et al. (2019) evaluated aerosol optical properties of an ensemble of models over Europe finding that differences due to diversity in modeling systems were larger than when using different emission inventories or when turning aerosol radiative feedbacks on and off. Reddington et al. (2016) evaluated a global aerosol model in tropical regions affected by biomass burning. They found that the model under-estimated AOD more than $PM_{2.5}$, even when an upper limit estimate

of aerosol hygroscopicity was assumed for the aerosols. Reddington et al. (2019) found further inconsistencies, as the model showed good representation of observed vertical profile while under-estimating AOD and hypothesized it was due to uncertainties in the AOD computations. In another study, Zieger et al. (2013) compared observations of scattering enhancement due to hygroscopic growth against results from the Optical Properties of Aerosols and Clouds (OPAC) software module showing a systematic over-prediction. This over-prediction could lead to mismatches between AOD and

$PM_{2.5}$ in models using this code. Curci et al. (2019) evaluated black carbon absorption for an ensemble of models over Europe and North America finding that biases were driven by the mixing state assumptions in the optical properties computations.

KORUS-AQ (KORea United States Air Quality) Campaign was an international cooperative air quality field study in South Korea that measured local and remote (e.g., anthropogenic, biomass burning, dust) sources of air pollution affecting the

95 Korean peninsula during May-June 2016. The objectives of the present study are to: 1) Evaluate one of the forecast system used to support flight planning during the mission;  2) Assess the degree of consistency between aerosol optical properties and mass concentrations for the forecasting and other configurations; and, 3) Explain the identified discrepancies. The results will provide guidance to future model development and we expect will motivate this type of analysis for other modeling systems and locations.

## 2. Methods

### 2.1 Regional modeling

Air quality forecasts were performed using the Weather Research and Forecasting model (Skamarock et al., 2008) coupled to Chemistry (WRF-Chem) (Grell et al., 2005) to support both KORUS-AQ flight planning and post-campaign analysis. The modeling domains are shown in Figure 1, with a regional domain of 20 km resolution, covering major source regions of transboundary pollutants affecting the Korean Peninsula: anthropogenic pollution from eastern China, dust from inner China and Mongolia, and wild fires from Siberia (Saide et al., 2014). A 4 km resolution domain was nested to cover the Korean Peninsula and surroundings at higher resolution. This inner domain encompassed the region where the KORUS-AQ flights were planned and was able to better resolve local sources. The forecasts were performed once daily and used meteorological initial and boundary conditions from National Centers for Environmental Prediction Global Forecast System (NCEP, 2007) and chemical boundary conditions from the Copernicus Atmosphere Monitoring Service (Inness et al., 2015). Initial conditions for gases and aerosols were obtained from the previous forecasting cycle. AOD data assimilation was implemented for the outer domain using data from low-earth orbiting (GMAO Neural Network retrieval) and geostationary satellites (Geostationary Ocean Color Imager retrievals; Choi et al., 2018; Choi et al., 2016; Lee et al., 2010) as described in Saide et al. (2014). To our knowledge, this was the first near-real time implementation of assimilating geostationary AOD. Each assimilation step modified aerosol mass keeping the species distribution and size of each bin constant. Thus, the assimilation had the potential to change the bin-aggregated composition and size distribution when size bins with different composition were scaled differently. Although there was no data assimilation performed on the inner domain, this domain was initialized 18 hours after the outer domain and thus was influenced by data assimilation through initial and boundary conditions. The forecast configuration was based on WRF-Chem version 3.6.1 with modifications. The aerosol and gas-chemistry packages corresponded to the 4-size bin Model for Simulating Aerosol Interactions and Chemistry (MOSAIC, Zaveri et al., 2008) and a simplified hydrocarbon chemical mechanism (Pfister et al., 2014), both selected to reduce computational costs compared to using the 8 size bin MOSAIC configuration and more complex chemical mechanisms. Although detailed secondary organic aerosol (SOA) formation schemes have been implemented for the 4-bin MOSAIC configuration (Shrivastava et al., 2013; Knote et al., 2015), it increases computational costs significantly. Thus, the simplified SOA formation scheme, proposed by Hodzic and Jimenez (2011) and verified to work well in multiple later studies (e.g., Hayes et al., 2015; Shah et al., 2019), was implemented to keep computational costs low. While this scheme included anthropogenic and biomass burning SOA, biogenic SOA was also modeled by using a SOA precursor surrogate derived from isoprene as described in Shrivastava et al. (2011). Aerosol-radiation interactions were included (Fast et al., 2006), while aerosol-cloud interactions were excluded to avoid the computational costs of tracking the cloud-borne aerosols. Anthropogenic emissions were developed by Konkuk University for KORUS-AQ forecasting and are described in Choi et al. (2019a) and Goldberg et al. (2019b). Natural dust, sea-spray and biogenic emissions were computed online using the Goddard Aerosol Radiation and Transport (GOCART) scheme (Ginoux et al., 2001; Zhao et al., 2010), following Gong et al.

(2002), and estimates from the Model of Emissions of Gases and Aerosol from Nature (MEGAN, Guenther et al., 2006), respectively. Biomass burning emission estimates were obtained from the Quick Fire Emissions Data set (Darmenov and da Silva, 2015) and were added using the online plume-rise model implemented in WRF-Chem (Grell et al., 2011). Other modeling configurations related to meteorological parametrizations and analysis nudging are described in Saide et al. (2014). In addition to the forecasting results, we performed retrospective sensitivity simulations summarized in Table 1. We first used the same configuration as the forecast, which we labeled MOSAIC4b. To explore the model sensitivity to increasing the resolution of the aerosol size bins, we performed simulations using the MOSAIC 8-bin configuration, coupled to the Carbon-Bond Mechanism version Z (CBM-Z) chemical scheme (Zaveri and Peters, 1999) and labelled it MOSAIC8b. Some caveats of this sensitivity simulation are that it uses a different gas-phase chemistry scheme and does not include secondary organic aerosol formation, thus this needs to be considered in the analysis when comparing it to the base configuration. WRF-Chem can also be configured with the Modal Aerosol Dynamics Model for Europe (MADE) model, where aerosol sizes are represented by log-normal modes (as opposed to sections as for MOSAIC). We used the configuration coupled to the updated Regional Atmospheric Chemistry Mechanism (RACM, Ahmadov et al., 2014) which contains secondary organic aerosol formation using the volatility basis-set (Ahmadov et al., 2012) and aerosol optical properties calculations (Tuccella et al., 2015). We label these simulations as MADE#, with # going from 1-4 depending on changes to parameters described in Table 1.

Simulations MOSAIC4b, MOSAIC8b and MADE1 are referred to as base configurations, while MADE2-4 are sensitivity simulations. All retrospective simulations were performed only for the 20 km resolution domain in this study, as we focus on a pollution event from long range transport (May 24-26 2016). Also, no data assimilation was performed for these simulations. Unless otherwise noted, they use the same inputs and parametrizations as for the forecast simulations.

## 2.2 Optical properties calculation

Aerosol optical properties in WRF-Chem are computed using a Mie code and Chebyshev expansion coefficients for each size bin, assuming an internal mixture within the bin and a volume mixing rule (Fast et al., 2006). The refractive indices (real part) and density used for each species are defined in Table 2 under the base configuration column. Only black carbon and other inorganics (OIN, inorganic aerosols other than sulfate, ammonium, nitrate, chloride and sodium) are considered to absorb solar radiation with an imaginary refractive index of 0.71 and 0.006, respectively. For the MOSAIC configurations, the size bins in the optical properties calculation correspond to those in the aerosol model (4 and 8 size bins, defined in Table 3); while, for modal aerosols, the modes are mapped to 8 sectional bins (same boundaries as the MOSAIC 8 bins) before the calculation by computing the aerosol mass and number concentration included in each section. WRF-Chem computes optical properties for ambient conditions. Thus, derivation of dry-extinction and scattering enhancements due to hygroscopic growth at fixed relative humidity requires computations at the post-processing stage. These computations require the aerosol water to be recalculated for the specified relative humidity. Since both the MOSAIC and MADE aerosol models compute aerosol water based on aerosol thermodynamics, versions of these computations are needed at the post-processing stage. In order to

simplify the process and add additional capabilities, an alternative optical properties code at the post-processing stage was developed that mimics the WRF-Chem one. This alternative approach uses Mie calculations from Mätzler (2002) code which is based on the Appendix of Bohren and Huffman (1983). Aerosol water uptake was parameterized using the method proposed by Petters and Kreidenweis (2007), which utilizes the hygroscopicity parameter ($\kappa$). Values for the hygroscopicity

parameter representing the base configuration are obtained from the WRF-Chem code used to compute aerosol water for the Goddard Chemistry Aerosol Radiation and Transport (GOCART, Chin et al., 2002) model which implements the $\kappa$ approach and a volume mixing rule. Following this configuration $\kappa$ is set to 0.5 for ammonium sulfate and ammonium nitrate. For sodium chloride $\kappa$ is set to 1.5 (Zieger et al., 2017). For organic aerosol, black carbon, and dust, $\kappa$ is set to zero as these aerosol types are currently not considered as electrolytes in the MOSAIC and MADE thermodynamic models and thus do

not contribute to water uptake in these frameworks (Fast et al., 2006). Ways to improve these simplifications will be discussed later in the text. Figure 2 shows an evaluation of the alternative approach against the WRF-Chem routines used in post-processing mode for the MOSAIC model that were developed as part of the data assimilation scheme (Saide et al., 2013). Under dry conditions, the alternative approach shows similar results as the WRF-Chem optical properties with the extinction to mass ratio curve, following each other for all sizes (Fig. 2a). The high frequency oscillations shown by the Mie

code of our alternative approach are smoothed in WRF-Chem due to the use of the Chebyshev expansion coefficients (Fast et al., 2006) and interpolation between wavelengths (optical properties at 400 nm and 600 nm are used to derive values at mid-visible wavelengths). Water uptake using the MOSAIC approach and the one described here provides similar results (Fig 2b), with values ~7% lower in the alternative approach that will be taken into consideration when evaluating the optical properties code in the next sections. The alternative optical properties code provides flexibility to evaluate changes in

configuration that would be difficult to implement in the WRF-Chem optical properties code. These include using more than 8 bins to improve size resolution, using variable density for aerosol species, and altering $\kappa$ to vary the extent that different aerosol chemical species take up water.

## 2.3 Airborne Observations

Airborne data used in this study were measured by instruments on board of the NASA DC-8 research aircraft as part of the

190 KORUS-AQ campaign (Aknan and Chen, 2019) during the flight starting at 22:00 UTC on May 24[th] (May 25[th] in local Korean time) 2016. This flight focused on measurements over the Yellow Sea and sampled some of the highest aerosol mass concentrations of the deployment originating from mostly anthropogenic pollution from China (Peterson et al., under review; Nault et al., 2018) . This flight was also chosen as it corresponds with a period of large model discrepancies (see Results Section). Measurements used in this study include numerous in-situ chemical composition and mass concentration and

195 physical properties of the aerosol, and remote sensing physical properties of the aerosol. $PM_1$, not including black carbon, was measured by the University of Colorado Boulder, High-Resolution Time-of-Flight Aerosol Mass Spectrometer (HR-ToF-AMS, hereinafter "AMS" for short; DeCarlo et al., 2006; Nault et al., 2018). These measurements included the mass concentrations of sulfate, nitrate, ammonium, chloride, and organic aerosol, and the estimated aerosol density. The

estimation of aerosol density is described in Nault et al. (2018) and DeCarlo et al. (2004). Refractory black carbon concentrations were measured by the NOAA Single Particle Soot Photometer (SP2; Lamb et al., 2018). Bulk water-soluble, inorganic aerosol was measured by the University of New Hampshire using Teflon filters, followed by off-line ion chromatography with an estimated aerodynamic diameter cut-off of ~4 µm (SAGA, Dibb et al., 2000; McNaughton et al., 2007). The in-situ, physical aerosol properties were measured by NASA Langley Aerosol Research Group (LARGE), which included dry aerosol scattering, extinction and single-scattering albedo, and scattering enhancements due to hygroscopic growth. These measurements were done with two TSI nephelometers (at 450, 550, and 700 nm wavelength) and a Particle Soot Absorption Photometer (at 470, 532, and 660nm wavelength) as described in Ziemba et al. (2013). Aerosol size distributions were also measured by the LARGE suite using a Scanning Mobility Particle Sizer (SMPS, TSI model 3936), a Laser Aerosol Spectrometer (LAS, TSI model 3340) and an Aerodynamic Particle Sizer (APS, TSI model 3321). AMS and SP2 measure mostly submicron aerosols while the LARGE inlet cut-off is at 5 µm aerodynamic diameter (McNaughton et al., 2007). Relative humidity was estimated using measurements of water vapor from the NASA Langley/Ames Diode Laser Hygrometer (Podolske et al., 2003). Finally, extinction curtains were measured using the Airborne Differential Absorption Lidar – High Spectral Resolution Lidar (DIAL–HSRL; Hair et al., 2008), from the NASA Langley LIDAR group. In-situ data was obtained from the 1 second merges (version 3) and merged with the corresponding version of data that was available through individual files (e.g., size distributions).

Though the instruments measuring size distributions overlap in some size bins, they use different sampling frequencies, and they use different sizing techniques based on different measures of aerosol diameters (e.g., geometric, optical and aerodynamic). Thus, these measurements need to be homogenized and combined to obtain a single size distribution. Thirty-two size bins using geometric diameter from a lower bound of 39 nm to an upper bound of 10 µm with a width (dlnD) of 0.1737 are used to re-bin the SMPS, LAS and APS size distributions. These boundaries and width are chosen so the distributions can be easily aggregated to the modeled size bins (Table 3). Data from the SMPS, LAS and APS are used for bins 1-8 (39 nm – 156 nm), 9-18 (156 nm- 743 nm) and 19-32 (743 nm – 10 µm), respectively. APS measures aerodynamic diameters, thus these are converted to geometric diameter by multiplying aerodynamic diameter by $\sqrt{X/\rho}$ (valid in the continuum regime that is where most of the coarse mode aerosol mass is in this study) assuming dynamic shape factor (X = 1.6) and density ($\rho$ = 2.6 g/cm$^3$) of dust aerosols, which we assume dominates the coarse mode aerosol. This assumption is made as AMS and SAGA measured similar concentrations of inorganic species (Figure 3a), and thus sulfate, ammonium, chloride and nitrate was mostly not present in sizes covered by SAGA but not by AMS. But since the aerosol size distribution measurements do show aerosol presence in these coarse sizes, we assume it is dominated by dust. Although LAS measures geometric diameter when particles are spherical, it is calibrated with NIST traceable polystyrene latex spheres (PSL), which have a larger refractive index (1.595) than the mixtures measured during the flight. Thus, LAS diameters are multiplied by 1.115 to approximately correct for this difference (Nault et al., 2018). While LAS and APS results are reported at 1 Hz frequency, SMPS provides data every minute. Since most datasets used in this work are provided at 1 Hz, we use

nearest neighbor interpolation to assign SMPS values at 1 Hz resolution. This is likely to have negligible impact in our results as there is little aerosol mass in the bins assigned from the SMPS and mass extinction efficiency is low at these sizes. Previous studies have found a saturation of the LAS detector for large aerosol number and mass concentrations (Liu et al., 2017; Nault et al., 2018), which occurs when scattering from individual particles start overlapping so that the signal does not go down to the baseline between events. This was the case for a large fraction of the measurements in the haze layer during the flight studied. Figure 3b shows that while this saturation is evident for large aerosol mass concentrations, for lower aerosol mass concentrations (where no saturation is expected) the LAS measures larger volume concentrations than the AMS+SP2 by ~11%. Although 11% falls within the stated accuracies of both measurements, it is also potentially reflecting a true difference in concentrations. The AMS detected exotic metals not typically reported, such as rubidium (Figure 4a), in the haze event. There was, on average, 10 ng sm$^{-3}$ rubidium in the plume between 01:30 to 05:00 UTC (and up to 71 ng/sm$^3$). Rubidium originates either from soil (e.g., dust; Kabata-Pendias and Pendias, 2001) or anthropogenic emissions, such as dust from steel and aluminum industries (Dillner et al., 2006; Tang et al., 2018). Rubidium is one of numerous types of metal emitted from these sources and would account for the minority of the mass for these emissions (Dillner et al., 2006). Also, the detection shown here is likely a lower limit of the actual concentrations considering the refractory nature of the aerosols typically containing rubidium. The presence of rubidium would suggest other inorganic material present in the haze event not typically measured by the AMS, suggesting the 11% difference in volume is due to these types of compounds. Thus, we corrected the LAS submicron number and volume distributions using the aerosol mass measured by the AMS+SP2 accounting for the ~11% volume not detected. For this, scaling factors were computed using the aerosol volume (estimated using the measured aerosol mass and the aerosol density reported by the AMS), corrected by 11%, and dividing by the measured LAS volume, accounting for the AMS transmission. The AMS transmission considers 100% and 0% efficiency at aerodynamic diameters of 550 nm and 1.5 µm, respectively, and a linear decrease in between using the logarithm of the aerodynamic diameters (Hu et al., 2017). The transmission curve was converted to geometric diameter for each observation using Equation 28 from DeCarlo et al. (2004) iteratively to update the Cunningham slip correction factor until convergence. The LAS correction assumes that the fractional contribution of aerosol not measured by AMS+SP2 is constant, which is a limitation of the approach, but we expect it to have limited impact in the analysis due to the small contribution. Submicron dust aerosol mass concentration is estimated using this volume residual, assuming dust density of 2.6 g/cm$^3$ (value used in WRF-Chem), and falls into the OIN aerosol category when comparing to model estimates.

**2.4. Ground based observations**

We also used multiple sources of ground-based observations. This includes Level 2.0 AOD at 500 nm wavelength, which was obtained from the Aerosol Robotic Network (AERONET; Holben et al., 1998) version 3 algorithm (Giles et al., 2019). During KORUS-AQ, the AERONET team enhanced the long term AERONET sites with a short term DRAGON network (Holben et al., 2018) to assess mesoscale spatial variability of aerosol properties, thus a total of 21 locations were available during the campaign period. We also used PM$_{2.5}$ and PM$_{10}$ from the air quality network maintained by Korean National

Institute of Environmental Research (NIER). For the period analyzed, PM$_{2.5}$ and PM$_{10}$ data were available from 320 and 329 locations, respectively, distributed across the peninsula.

## 3. Results and discussion

### 3.1 Forecast evaluation

Figure 5a shows comparisons of AOD measured by the AERONET network over South Korea versus forecasted AOD at the
site locations. The model shows good performance over the period (e.g., mean AOD for observations and model are 0.58 and 0.60, respectively). This performance is expected as the system is assimilating satellite AOD, and satellite AOD retrievals have shown good agreement with AERONET data in the region (Choi et al., 2019b). However, the forecasts generally show large over-predictions of surface particulate matter for the period of large concentrations in the peninsula (mean bias for PM2.5 and PM10 is 44 µg/m$^3$ and 21 µg/m$^3$, respectively), consistent with previously reported results (Lennartson et al.,
2018). These over-predictions are more severe for PM$_{2.5}$ during the passing of the transboundary pollution coming from China (May 25$^{th}$ and 26$^{th}$), sometimes exceeding a factor of two difference. This points towards model deficiencies in connecting surface mass concentrations with column optical properties, which is explored in this study focusing on the day the aircraft sampled this airmass (May 24$^{th}$). Also, note that PM$_{2.5}$ and PM$_{10}$ are similar in the model while PM$_{10}$ is larger than PM$_{2.5}$ in the observations (reflected in the differences in mean bias), pointing towards model biases in representing
coarse mode aerosols.

Figure 6 shows observed and forecasted AOD retrievals at noon local time the day of the May 24$^{th}$ DC-8 flight that sampled the transboundary pollution that affected the Korean peninsula. The forecasts shown have not yet assimilated the AOD retrieved that day, showing the ability of the system to carry forward the information assimilated the day before. AODs larger than 1 were found over areas in the Yellow Sea that were correctly forecasted and that enabled the KORUS-AQ team
to plan a successful flight (track in yellow on the right panel of Figure 6) in the region. Data from this flight are used to perform a detailed model evaluation to understand the model biases.

One potential reason for the discrepancies found could be related to model representation of aerosol vertical profiles. Figure 7 shows aerosol extinction curtains over the DC-8 trajectory (aircraft altitude in red solid line), as sampled by the DIAL-HSRL, and as predicted by the forecasts. The haze layer is mostly confined to below 2 km, which the model represents
properly (e.g., mean extinction in this layer for observation and model is 0.44 1/km and 0.38 1/km, respectively). The model has slightly higher mixing layers which, if anything, would lead to opposite biases (e.g., under-estimation of surface concentrations). A layer with lower aerosol extinction found between 2 and 6 km, which DIAL-HSRL classified as dust (not shown) and where SAGA reported elevated levels of Ca$^{+2}$ associated to dust, is also well captured by the model in terms of both aerosol type and amount (mean extinction in this layer during 00-01 UTC for observation and model is 0.012 1/km and
0.015 1/km, respectively). Thus, we discard issues with model representation of the vertical placement of the plume as reasons for explaining the AOD to PM inconsistencies mentioned earlier.

AOD is also highly sensitive to relative humidity (Brock et al., 2016a) and previous studies have explained AOD biases due to model issues representing relative humidity (Feng et al., 2016). In-situ measurements of relative humidity in the haze layer showed average values of 62% with an interquartile range of 9% (57-66 %). The forecast shows a reasonable representation with slightly higher average value (64%) and interquartile range (10%). Thus, we conclude skill in predicting relative humidity is not related to the discrepancies found in this study.

From this analysis, we conclude that the likely reason for the discrepancy resides in the computation of the aerosol optical properties, i.e., how aerosol mass is translated to AOD, and thus in the following sections we perform a thorough evaluation of this topic using in-situ airborne data.

## 3.2 Closure studies

Model representation of optical properties can be separated into two stages: 1) how well the model represents the aerosol properties which drive the optical properties computation (e.g., size distribution, composition, concentrations, etc.); and, 2) the accuracy of the optical properties code. The latter can be evaluated by driving the optical properties code using observed quantities and comparing the outputs with measurements of aerosol optical properties, a methodology that has been applied for previous field campaigns (Barnard et al., 2010; Brock et al., 2016a) and that we will refer to as "closure study" here. This allows us to isolate issues regarding the optical properties code and to assess ways to improve the model representation of optical properties.

One challenge of closure studies is that the optical properties code requires speciated and size-resolved aerosols, thus assumptions need to be made on how to distribute the measured chemical species into the size bins. Figure 3a shows a scatter plot of SAGA filter-based vs AMS online measurements of inorganic aerosol mass concentrations (sulfate, nitrate, ammonium, chloride), showing good agreement for this flight. While secondary inorganic aerosol mass concentrations were elevated in the coarse mode for other KORUS-AQ flights (and thus not detected by AMS; Heim et al., 2020), for the flight analyzed here this fraction seems to be negligible. Thus, we assume that the tail of the coarse mode is composed of only OIN (likely dust). We also assume that composition is not size dependent within the accumulation mode. Size resolved AMS measurements support this assumption by showing similar composition within the accumulation mode (Fig 4b). We set the aerosol diameter cut-off between the accumulation mode and the lower tail of the coarse mode at 884 nm, based on size distribution measurements (Fig. 8). Also, since both LAS and APS cover the lower tail of the coarse mode we use APS estimates in this range because LAS presents lower volume concentrations.

Figure 9 shows multiple statistical metrics in the form of box and whisker plots for observations and closure results during the three consecutive hours that the DC-8 spent measuring the haze layer at multiple altitudes (2:00-5:00 UTC, see Figure 7). The different closure scenarios are described in Table 4 and consist of the base configuration and then the base with varying parameters such as the size bin resolution, refractive indices, and hygroscopicity parameter.

### 3.2.1 Dry extinction

A variable that is typically computed to assess the efficiency of an aerosol population at scattering light is the ratio between dry scattering/extinction and aerosol mass concentrations, which is typically referred to as "mass scattering/extinction efficiency". Note that for this study, aerosol mass concentration corresponds to that measured by AMS+SP2. We also define "volume extinction efficiency" as the ratio between dry extinction and aerosol volume concentration, obtained from the aerosol size distribution measurements after performing the corrections described in section 2.3. As AMS and SP2 measure mostly submicron aerosols of select chemical species, there is potential for unaccounted aerosol mass that is contributing to aerosol extinction that could complicate the interpretation of the mass extinction efficiency. Therefore, the volume extinction efficiency is reported in addition to the mass extinction efficiency as the extinction and volume are measured for all aerosols in the same size range (behind the LARGE inlet).

Figure 9a,b clearly show how the base configuration of the optical properties code drastically under-predicts the mass and volume extinction efficiencies (e.g., mean mass extinction efficiency for observations and Closure 1 is 6.7 $m^2/g$ and 4.5 $m^2/g$, respectively), consistent with the discrepancies shown in Figure 5. Aerosols are binned into 4 size bins in the base configuration (Closure 1); thus, Closure 2 and 3 explore finer binning to 8 and 16 sections, respectively. The finer binning does improve the performance, especially when going from 4 to 8 bins (average mass extinction efficiencies of 4.5 $m^2/g$ and 5.0 $m^2/g$, respectively). Figure 8 shows the size distributions for the three types of size aggregation, showing a large diversity in the bin concentrations contributing to the total mass in fine-resolution bins, which is lost when aggregating to coarser-resolution bins. Figure 10 shows steep changes in the volume extinction efficiency in the diameters where most of the aerosol mass is found (200-500 nm). In the 4-bin configuration, the whole accumulation mode is included in one bin. After aggregation, a mean diameter of 293 nm is obtained, which has a volume extinction efficiency below 6 $m^2/m^3$ with base refractive indices. On the other hand, a large percentage of the accumulation mode is found in bin #4 with the 8-bin configuration, which has a mean diameter of 380 nm and a volume extinction efficiency of ~8 $m^2/m^3$, which raises the overall efficiency substantially. The improvements from 8 to 16 bins are lower than from 4 to 8 bins, but still significant, and are due to similar reasons. For instance, bin #8 in the 16-bin configuration shows volume extinction efficiency above 9 $m^2/m^3$, getting close to the maximum values for base refractive indices. Negligible improvements are found when further refining from 16 to 32 size bins (not shown).

Although improvements are found when refining the size bins, significant biases still persist for the 16 bin configuration (Fig. 9a,b). Thus, we explore modifying the refractive indices used in the Mie calculation (see section 2.3), based on values reported in the literature for the aerosol species accounting for most submicron mass. Typical real refractive indices assumed in closure studies (e.g., Brock et al., 2016a) for ammonium-sulfate and ammonium-nitrate are 1.527 (Hand and Kreidenweis, 2002) and 1.553 (Tang, 1996) which are larger than those used in the base configuration, thus we update them accordingly. For primary and secondary organic aerosols, there is a large range of values found in the literature (e.g., Moise et al., 2015; Lu et al., 2015). Aldhaif et al. (2018) derived OA real refractive index from field deployments by air mass type, finding a

mean value of 1.54 with 1.52-1.55 as the 25-75[th] percentile range for urban air masses. We chose the value of 1.55 as it is in the 25-75[th] percentile range and because it is a typical value used in past studies (Zhang et al., 1994; Hand and Kreidenweis, 2002; Hodzic et al., 2004). This value also corresponds to the mean real refractive index reported by Lu et al. (2015) for primary organic aerosol based on a literature review. The Closure 4 case includes these updates (summary of updated parameters in Table 2) showing an increase in the efficiencies which improves the model representation (Figure 9a,b). Although the mass and volume extinction efficiencies are still under-predicted (e.g., average mass extinction efficiencies of 6.7 m$^2$/g and 6.1 m$^2$/g for observation and Closure 4, respectively), there is much better agreement when using the updated refractive indices, obtaining an overlap of the observed and modeled 25-75[th] percentile boxes. Besides the overall increase in the efficiencies, there is also a slight shift towards smaller sizes of the location where the efficiencies vs particle dry-diameter curve achieves its maximum (Figure 10a). The update in the OA refractive index generates the most impact (not shown) due to the larger increase (7% change vs 0.5% and 3.5% for ammonium sulfate and ammonium nitrate, respectively) and large contribution to the total mass (23% on average).

### 3.2.2 Hygroscopic growth

While the analysis in the previous sub-section was performed for the dry aerosol extinction, we also explored possible biases due to hygroscopic growth, considering that relative humidity was in the 50-80% range in the haze layer. We assessed the performance of the optical properties code, driven by observed inputs in representing the aerosol light scattering enhancement factor (f(RH)), defined here as the ratio between 550 nm aerosol scattering at 80% (wet) and 20% (dry) relative humidity.

Figure 9c shows that the base configuration performs well for f(RH) (average of 2.2 vs 2.1 for observation and Closure 1, respectively). This is due to a combination of the wrong reasons (i.e., cancellation of errors), as it deteriorates when increasing the size bin resolution (Closure 2 and 3) and upon increasing the refractive indices (Closure 4), going down to average values as low as 1.7. Figure 10a additionally shows f(RH) and can help explain this behavior, as f(RH) has a strong decreasing trend with increasing diameter in the region <350 nm. Thus, because the 4-bin representation displays an apparent decrease in the mean diameter (Figure 7), f(RH) is over-estimated. As the size bins are refined, less aerosol mass falls in the smaller size bins, decreasing the total f(RH). The further decrease of f(RH) with increasing refractive indices at these size ranges is also shown in Figure 10a. While our alternative approach at computing aerosol water uptake resulted in values ~7% lower than that shown by WRF-Chem, the difference in the observed vs Closure4 f(RH) are close to 30%, thus we conclude that similar biases would be expected for the WRF-Chem routines.

To improve the optical properties code performance, we updated the hygroscopicity parameter based on values found in the literature for the species contributing to most of the aerosol mass. κ values are generally reported with large range of uncertainty and can depend on the measurement technique and environmental conditions. Petters and Kreidenweis (2007) show a wide range of κ values for ammonium sulfate, from 0.33 to 0.72 and a mean of 0.53, for κ derived using growth factors, and a mean value of 0.61 based on CCN measurements. For ammonium nitrate, only CCN derived κ is available

with a mean value of 0.67 and a range of 0.577-0.753. We chose to use the mean values of the CCN-derived estimates (0.61 for ammonium sulfate and 0.67 for ammonium nitrate), as they are contained in the ranges provided in Petters and Kreidenweis (2007) and in other studies (e.g., Good et al., 2010). For organic aerosol, the range is even larger. We chose to treat organic aerosol and OIN as slightly hygroscopic, as it was originally specified in WRF-Chem parametrization for GOCART, with κ values of 0.14 for both, which is consistent with values reported for aged urban OA and for rural environments (Wang et al., 2010; Mei et al., 2013; Levin et al., 2014). κ of sodium chloride was updated to 1.1 following the revisions of Zieger et al. (2017) to consider the properties of inorganic sea salt. This large decrease in κ has little impact for the study period as sea salt mass concentrations represented less than 1% of the total on both observations and models. A summary of the updated κ values can be found on Table 2. Figure 9c shows significant increases in f(RH) from Closure 4 to Closure 5 (average f(RH) of 2.1) up to a similar level as the observations. Sensitivity analysis shows that most of the change is related to the κ increases in ammonium sulfate and ammonium nitrate due to their larger contribution to mass fraction (62% on average), while additional water uptake of organics and other inorganic aerosols play a minor role. Thus, choosing a lower OA κ value more consistent with other studies (Brock et al., 2016a; Shingler et al., 2016) would have resulted in similar findings.

### 3.2.3 Other aerosol optical properties

Figures 9d and 9e show the change for the Ångström exponent (AE) and single-scattering albedo (SSA) for the different model closure configurations shown. Overall, the representation of both AE and SSA improve when going from the coarse size bin resolution and base parameters to the finer bin and updated parameters, which represents independent pieces of evidence that the change in configuration is in the right direction. As seen in Figure 10b, AE is sensitive to the aerosol size distribution and generally decreases with larger aerosol sizes. This explains the sharp decrease when refining the aerosol size bins (mean AE drops from 2.4 in Closure 1 to 2.2 in Closure 2), due to the lower mean diameters for the 4-bin configuration as described previously. Figure 9e shows that SSA gradually increases with the change of configuration (from 0.92 in Closure 1 to 0.94 in Closure 5), which is due to changes in mean diameters and higher scattering when increasing the real refractive indices (Fig 10b). While the AE of Closure 5 matches very well with the observed values (mostly due to improvements from Closure 1 to 2), SSA is still slightly under-predicted (mean observation of 0.95), which is an issue previously identified in other closure studies using a similar approach for computing aerosol optical properties (Barnard et al., 2010). This under-estimation could be due to multiple uncertainties including assumptions on black carbon and OIN complex refractive indices, black carbon mixing state, and size-independent black carbon fractional contribution to the accumulation mode.

### 3.3 Evaluation of retrospective simulations

The previous section provides clarity on what we should expect from the optical properties code if the model was reproducing observed aerosol size distributions and composition. In this section we perform a similar analysis but driving the

optical properties code with simulated aerosol properties (summarized in Table 1 and shown in Figure 11). Comparing Figures 11a and 11b, we can see large discrepancies in the performance of the mass and volume extinction efficiencies, opposite to the closure studies (Figure 9) where they remain consistent. For instance, MOSAIC4b shows good representation for the mass extinction efficiency against Closure 5 (mean of 6.4 $m^2/g$ and 6.2 $m^2/g$, respectively) but largely under-estimates the volume extinction efficiency (mean of 4.5 $m^2/m^3$ and 7.5 $m^2/m^3$, respectively). Comparing Figure 11 and 9, the mass extinction efficiency is shifted up in the three base modeling configurations, and thus a refinement in size bins (going from MOSAIC4bin to MOSAIC8b or MADE1) has the opposite effect on performance for mass and volume extinction efficiencies. The modeled aerosol mass concentration used when computing the mass extinction efficiency is that for which the AMS transmission is applied. Thus, one possible explanation could be that the models have significant aerosol mass outside of the sizes where the AMS can detect. This would reduce the mass concentration after applying the transmission curve which would increase the extinction to mass ratio (i.e., the mass extinction efficiency) due to the unaccounted mass that contributes to extinction. Figure 12 shows the size distributions and AMS transmission efficiency for the simulations, where this issue is evident as all three base model configurations (MOSAIC4b, MOSAIC8b and MADE1) place substantial aerosol accumulation mode mass in sizes where the AMS transmission starts decreasing (> 625 nm). This is not the case for the observed size distribution (Fig. 8), where most accumulation mode aerosol mass is within the AMS transmission and explains the consistency between mass and volume extinction efficiency for closure studies.

Another contributor to this discrepancy is the model prediction of chemical composition. Figure 13 shows that, although OIN absolute concentrations are in the range of the observations, all modeling configurations over-predict the fractional contribution of submicron OIN mass (17-28% vs 12% in the observations), with the MOSAIC configurations showing larger over-predictions. Since the aerosol mass used in the mass extinction efficiency corresponds to that measured by AMS+SP2 where OIN is not included, then over-predicting the OIN fraction would increase the mass not accounted for in the ratio, increasing it relative to the observations. Potential reasons contributing to the over-prediction in OIN fraction include: (1) an over-prediction of the "other PM2.5" anthropogenic emission category and/or distributing it in the accumulation mode as opposed to in the lower tail of the coarse mode; (2) over-prediction of fine mode by wind-blown dust parameterizations (Kok, 2011); and/or, (3) insufficient production of secondary organic and inorganic aerosols (see under-prediction on left panel of Fig. 13), which has the effect of increasing the fractional contribution of the primary aerosol species (OIN in this case). Under-predictions of organic aerosol could be explained by large variations of secondary organic aerosol production within urban areas (Nault et al., 2018) that are not captured by the modeling configurations. Under-predictions of secondary inorganics could be due to missing mechanisms to produce sulfate during Chinese haze conditions (e.g., Gao et al., 2016a). These mechanisms were not included in this study as uncertainties remain on the actual pathways (Guo et al., 2017) and representation in models (Song et al., 2018). Other potential reasons for the under-estimate on secondary inorganics could include slow in-cloud $H_2O_2$ oxidation of $SO_2$ due to under-estimates of cloud volume and $NO_x$ underpredictions (e.g., Goldberg et al., 2019b).

Another point to note is that models under-predict the relative magnitude of the coarse aerosols (2.5-10 μm range, bin #4 in the 4-bin configuration). This helps to explain why the biases shown in Figure 5 are more pronounced for $PM_{2.5}$ than $PM_{10}$, as the under-prediction in the coarse aerosols is offset by the over-prediction in the fine aerosols, and is consistent with findings from previous studies (Balzarini et al., 2015; Im et al., 2015).

As mentioned earlier, all three base modeling configurations have issues representing the size distribution regardless of the large diversity in chemical and aerosol scheme. This is a topic that needs to be explored further in a future dedicated study. In the case of the MOSAIC configurations, the shape of the size distribution evolves through aerosol processes (coagulation, condensation, etc.). Since these processes that modify the size distribution are reasonably well known (Seinfeld and Pandis, 2016), it is unlikely such large errors would arise from the model implementation of these processes. A more likely explanation is that the shape of the size distribution established at the point of emission is too wide to start with and unfolds into the results shown. In the case of the MADE1 configuration, the widths of the log-normal modes are controlled by the geometric standard deviation (GSD). In the WRF-Chem implementation the GSD is fixed at 1.7, 2.0, and 2.5 for nuclei, accumulation and coarse modes, respectively. Changing the GSD to 1.6 for both the nuclei and accumulation mode (i.e., MADE2 simulations) results in a better representation of the observed aerosol size distribution (Fig. 12 vs Fig. 8), with a narrower accumulation mode peaking in the 300-450 nm range and much smaller mass contribution in sizes above 625 nm. These results are consistent with Brock et al. (2016b), which showed that GSD in the southeastern US are in the 1.4-1.6 range while chemistry-climate models generally over-predict it by using a GSD value of 2.0.

After correcting the modeled size distribution, a larger percentage of the aerosol mass is found within the AMS transmission (see increase in mass for all species from MADE1 to MADE2 in left panel of Fig. 13). Also, the model representation of mass and volume extinction efficiency against observations now follows the same trend for MADE2 (Figure 11a,b). Another reason for potential discrepancies is related to the aerosol density used for organic aerosols (OA) in the model. For the closure study, OA mass is converted to volume using the density reported by the AMS, which varies substantially with the oxidation state of organic aerosol (Kuwata et al., 2012). For the period analyzed here, the OA density has a mean of 1.5 $g/cm^3$ and 25th and 75th percentiles of 1.35 $g/cm^3$ and 1.6 $g/cm^3$, respectively. In the case of the simulations, the aerosol optical properties code in WRF-Chem uses a constant OA density of 1.0 $g/cm^3$. Thus, a lower density translates into larger volume per unit mass, increasing the mass extinction efficiency and explaining the remaining discrepancy. On the other hand, volume extinction efficiency is less sensitive to changes in aerosol density as a decrease in density decreases both extinction and volume. In fact, the volume extinction efficiency remains consistent with the analyses shown in the previous section when changing size distribution and aerosol density. Thus, we use volume extinction efficiency in the following analysis.

As shown in Figure 11b, the dry extinction to volume ratio is greatly under-estimated by almost a factor of two by the MOSAIC4b simulation, which helps explain the discrepancy described in Figure 5. As described in the previous section, large improvements are found when computing optical properties using finer aerosol bin representation with some remaining biases (MOSAIC8b and MADE1 bring mean volume extinction efficiency to 6.4 $m^2/m^3$ and 6.5 $m^2/m^3$, respectively). This

large improvement from the 4 to 8 bin configuration is in agreement with previous studies that found an overall consistency between AOD and surface PM when using 8 size bins (Saide et al., 2014; Gao et al., 2016b; Gao et al., 2015). Surprisingly, simulations with better aerosol size distribution representation (MADE2) do not significantly modify the representation of the volume and mass extinction efficiency (mean volume extinction efficiency remains at 6.5 $m^2/m^3$). Figures 12 and 10 show that both size distributions (MADE1 and MADE2 for the 8-bin configuration) are centered in a size range with high mass extinction efficiency (~9 $m^2/g$ around 400 nm diameter). While the original size distribution has less mass assigned to this high efficiency bin, it has substantial mass in larger size ranges where mass extinction efficiency is still high (~7 $m^2/g$ at ~800 nm diameter). On the other hand, the updated size distribution has larger mass in smaller size bins where mass extinction efficiency drops substantially (~3-4 $m^2/g$ at 200-250 nm diameter). Thus, the overall mass extinction efficiency remains similar for both size distribution due to compensating effects. As performed in the previous section, a sensitivity simulation is carried out such that the refractive indices of selected species are increased (MADE3,4), which bring the ratios to similar levels as the Closure 5 (mean volume extinction efficiency of 7.6 $m^2/m^3$ for MADE4 vs 7.5 $m^2/m^3$ for Closure 5).

In terms of aerosol hygroscopic growth and its effects on scattering, all base simulations under-predict f(RH) (1.7-1.8 on average, with observations and Closure 5 showing 2.2 and 2.1, respectively), which also helps explain the discrepancies shown in Figure 5. Improving the size distribution (MADE2) has a small but positive effect on f(RH) representation. The largest improvement is found when updating the hygroscopicity parameters for the same species updated in Closure 5 (average f(RH) of 2.0 for MADE4). After this, a slight under-prediction is still found. A possible contribution to this bias could be linked to the simulations not representing the aerosol chemical composition properly. As seen in Figure 13, MADE2 reasonably represents the observed pie-chart but does show a slight over-prediction of the less hydrophilic species (sum of OA, BC and OIN is 44% vs 37% in the observations). Another contributor to this bias is the low OA density used in the retrospective simulations. Using a lower aerosol density has a similar effect as using larger refractive index (more extinction per unit mass) and, as seen in Figure 10a, increasing the refractive index reduces f(RH) explaining the bias. Sensitivity analysis using observed OA density confirms this finding (not shown).

AE performance improves drastically when the size resolution is improved (average AE increases from 1.0 in MADE1 to 1.9 in MADE2) as the size distribution is shifted to smaller sizes, increasing AE. The AE after all the updates (MADE4) is still low (mean AE for MADE4 is 1.6 vs 2.2 in the observations), which is partially related to the low OA density but also might be associated with the modeled size distribution. Out of the base simulations, SSA performance is better for the MADE1 simulation (mean of 0.93) as MOSAIC8b (mean of 0.89) over-predicts the black carbon fraction (Fig. 13) and MOSAIC4b (mean of 0.90) has a coarse size bin representation (see previous section). After the updates (MADE4), SSA skill is comparable to that of the Closure 5 study (mean of 0.94)  likely due to BC being well represented both in magnitude and fractional contribution (MADE2 results shown in Figure 13 are similar to those from MADE4).

## 4. Conclusions

In this study, we first evaluated WRF-Chem forecasts, which included assimilation of AOD, performed to support flight planning during the NASA/NIER KORUS-AQ field campaign. While forecasts showed accurate predictions of aerosol optical depth, there were over-predictions of surface particulate matter in the Korean peninsula, with the largest deviations occurring for $PM_{2.5}$ during a transboundary pollution event. Additional analysis showed that the model was able to capture the vertical extent and the relative humidity of the haze layer, pointing towards issues related to the aerosol mass to optical properties calculation.

Further analysis was split in two sections. First, a closure study was performed by driving the optical properties parametrization with in-situ observations of aerosol size distributions and composition collected by the DC-8 aircraft. These were compared to measured optical properties, including mass and volume extinction efficiencies, hygroscopic growth represented by f(RH), Ångström exponent, and single-scattering albedo (SSA). This analysis showed closure was not possible by the base configuration and that multiple modifications were needed to achieve closure. These included driving the optical properties code with finer size bin representation (from 4 up to 16 bins), increasing refractive indices of ammonium nitrate (to 1.553) and organic aerosol (to 1.55) according to ranges found in the recent literature, and increasing hygroscopicity parameter of ammonium sulfate (to 0.61), ammonium nitrate (to 0.67), organic aerosol (to 0.14), and other inorganics (to 0.14) within published ranges. The coarse bin representation and low values of refractive indices and hygroscopicity parameters explain why the forecasts showed the largest discrepancies during the haze event, as these events are associated with large relative humidity and aerosol size distributions that peak close to the maximum mass extinction efficiencies.

Second, aerosol optical and microphysical properties were evaluated for retrospective simulations using three different aerosol models within WRF-Chem. This exercise additionally found that all three aerosol models were unable to properly capture the aerosol size distribution, showing a larger size range than what was observed. As a result, a substantial fraction of modeled aerosol mass was in sizes where the AMS transmission starts decreasing, which led to discrepancies between the modeled mass and volume extinction efficiencies. We also found that, while the model uses a value of 1.0 $g/cm^3$ for organic aerosol density, larger values were observed with a mean of ~1.5 $g/cm^3$ and considerable variability, which generated further discrepancies and reduced the skill in predicting some of the optical properties. Other issues included an over-prediction of the OIN fractional contribution by all models (which could be due to issues with OIN emissions and/or secondary aerosol formation). The size distribution of a configuration using a modal scheme improved by reducing the geometric standard deviation (GSD) of the accumulation mode from 2.0 to 1.6. Further increasing the refractive indices and hygroscopicity parameters (as noted above) provided overall better representation of optical properties. Future work needs to assess if the simulated size distributions can be improved by including primary aerosol emissions in the model using size-resolved and source-specific observational datasets (Winijkul et al., 2015; Lu et al., 2015).

A series of assumptions were made in this study that should be considered when analyzing the results. One of them corresponds to the correction of the LAS measurements to account for the PSL calibration and for the saturation of the instrument at high aerosol concentrations. The calibration correction was performed by applying a single scaling factor to the measured diameters which may not be constant due to changes in aerosol size and composition (Kupc et al., 2018). In addition, the saturation correction was applied for all LAS size bins equally, while the saturation of the instrument is both a function of particle concentration and size. The use of these assumptions could impact the results which could account for some of the minor mismatches found between observations and the closure study after accounting for possible model uncertainties (Closure 5). The LAS was the only instrument on board sampling the overall aerosol size distribution over the ranges where the accumulation mode peaked, so it would be useful for future field campaigns to have overlapping instruments with similar capabilities over this size range or dilute the sample during high concentration events as has been done in other deployments (Brock et al., 2019).

Consistency of relationships between AOD and PM in models is a key element in effectively improving predictions through data assimilation of AOD and/or PM mass. This work found that multiple sources of model uncertainties need to be addressed to provide accurate representation of optical properties and avoid mismatches when performing data assimilation driven by AOD observations. These include the use of fine aerosol size representation in optical properties calculations and improved representation of aerosol properties (size distribution, chemical composition, refractive index, hygroscopicity parameter, density). Accurate representation of aerosol optical properties is also important for other fields that use these models to make the connection between aerosol mass concentrations and aerosol optical properties, including assessments of aerosol health effects based on satellite data, proper projections of aerosol-radiation interactions done by climate models, visibility forecasts, and solar power predictions for energy applications.

In this study we evaluated different configuration of the WRF-Chem model for the specific case of anthropogenic outflow from China, thus future studies can perform similar analysis for other types of air masses and assess if the model configuration updates suggested here produce better results in other scenarios. Also, similar analysis is also needed for other air quality and chemistry-climate models to assess if similar biases or different ones arise.

**Code and data availability**

The WRF-Chem code and satellite AOD retrievals used in this study is available upon request. Korean air quality data can be found in http://www.airkorea.or.kr, AERONET data is available at https://aeronet.gsfc.nasa.gov/, and KORUS-AQ data and flight reports are public at https://www-air.larc.nasa.gov/missions/korus-aq/. Contact P.E. Saide (saide@atmos.ucla.edu) for data requests.

## Author contributions

PES designed and executed the study and led the writing of the manuscript. AB, CAC, KLT, BA, JWH, ARN, GSD, JLJ, BAN, PCJ, JD, EH, KL. JPS, SEP, JK, MC, and BH provided observational data. GP and AH provided code. All co-authors contributed with feedback during the development and writing of the study.

## Competing interests

The authors declare that they have no conflict of interest.

## Acknowledgments

We thank all KORUS-AQ participants that made the field experiment possible. We also thank the PIs and their staff of all the AERONET sites in South Korea for establishing and maintaining them. This work was carried out with the aid of NASA grants NNX11AI52G, NNX16AD96G, NNX15AT96G, NNX15AT88G, NNH15AB60I, and 80NSSC19K0124. Its contents are solely the responsibility of the authors and do not necessarily represent the official views of the funding institutions. The National Center for Atmospheric Research is sponsored by the National Science Foundation. JK and MC was supported by the 'Technology development for Practical Applications of Multi-Satellite data to maritime issues' funded by the Ministry of Ocean and Fisheries, Korea.

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

**Tables**

**Table 1. Summary of WRF-Chem simulations. Refractive indices, hygroscopicity parameters and size bins are defined in Table 2 and 3. Refer to section 3.2 and 3.3 for definitions on base and updated configurations.**

| Name | Aerosol Scheme | Size bins used for OP | GSD of the modes | Refractive index | Hygroscopicity |
|------|----------------|----------------------|------------------|------------------|----------------|
| MOSAIC4b | Sectional 4 bins | 4 bins | - | Base | Base |
| MOSAIC8b | Sectional 8 bins | 8 bins | - | Base | Base |
| MADE1 | Modal | 8 bins | Base | Base | Base |
| MADE2 | Modal | 8 bins | Updated | Base | Base |
| MADE3 | Modal | 8 bins | Updated | Updated | Base |
| MADE4 | Modal | 8 bins | Updated | Updated | Updated |

**Table 2. Real refractive indexes [dimensionless], hygroscopicity parameter [dimensionless] and aerosol density [g/cm³] used in the base and updated configurations. Values that changed in the updated configurations are noted in cursive. Note that measured**
**organic aerosol density was used in the closure studies.**

| | Real Refractive Index | | Hygroscopicity parameter | | Density |
|---|---|---|---|---|---|
| | base | updated | base | updated | base |
| ammonium sulfate | 1.52 | *1.527* | 0.5 | *0.61* | 1.8 |
| ammonium nitrate | 1.5 | *1.553* | 0.5 | *0.67* | 1.8 |
| sodium chloride | 1.45 | 1.45 | 1.5 | 1.1 | 2.2 |
| other inorganics | 1.55 | 1.55 | 0 | *0.14* | 2.6 |
| organic aerosol | 1.45 | *1.55* | 0 | *0.14* | 1 |
| black carbon | 1.85 | 1.85 | 0 | 0 | 1 |
| aerosol water | 1.33 | 1.33 | - | - | 1 |

**Table 3. Lower and upper diameters in μm for the 4, 8 and 16 size bins configurations**

| 4 bin | Lower | Upper |
|---|---|---|
| Bin 1 | 0.039 | 0.156 |
| Bin 2 | 0.156 | 0.625 |
| Bin 3 | 0.625 | 2.5 |
| Bin 4 | 2.5 | 10 |

| 8 bin | Lower | Upper |
|---|---|---|
| Bin 1 | 0.039 | 0.078 |
| Bin 2 | 0.078 | 0.156 |
| Bin 3 | 0.156 | 0.312 |
| Bin 4 | 0.312 | 0.625 |
| Bin 5 | 0.625 | 1.25 |
| Bin 6 | 1.25 | 2.5 |
| Bin 7 | 2.5 | 5 |
| Bin 8 | 5 | 10 |

| 16 bin | Lower | Upper |
|---|---|---|
| Bin 1 | 0.039 | 0.0552 |
| Bin 2 | 0.0552 | 0.078 |
| Bin 3 | 0.078 | 0.11 |
| Bin 4 | 0.11 | 0.156 |
| Bin 5 | 0.156 | 0.221 |
| Bin 6 | 0.221 | 0.312 |
| Bin 7 | 0.312 | 0.442 |
| Bin 8 | 0.442 | 0.625 |
| Bin 9 | 0.625 | 0.884 |
| Bin 10 | 0.884 | 1.25 |
| Bin 11 | 1.25 | 1.77 |
| Bin 12 | 1.77 | 2.5 |
| Bin 13 | 2.5 | 3.54 |
| Bin 14 | 3.54 | 5 |
| Bin 15 | 5 | 7.07 |
| Bin 16 | 7.07 | 10 |

**Table 4. Description of closure cases. Refer to section 3.2 for definitions on base and updated configurations. Size bins are defined in Table 3.**

| Name | # of size bins | Refractive index | Hygroscopicity |
|---|---|---|---|
| Closure1 | 4 | Base | Base |
| Closure2 | 8 | Base | Base |
| Closure3 | 16 | Base | Base |
| Closure4 | 16 | Updated | Base |
| Closure5 | 16 | Updated | Updated |

**Figures**

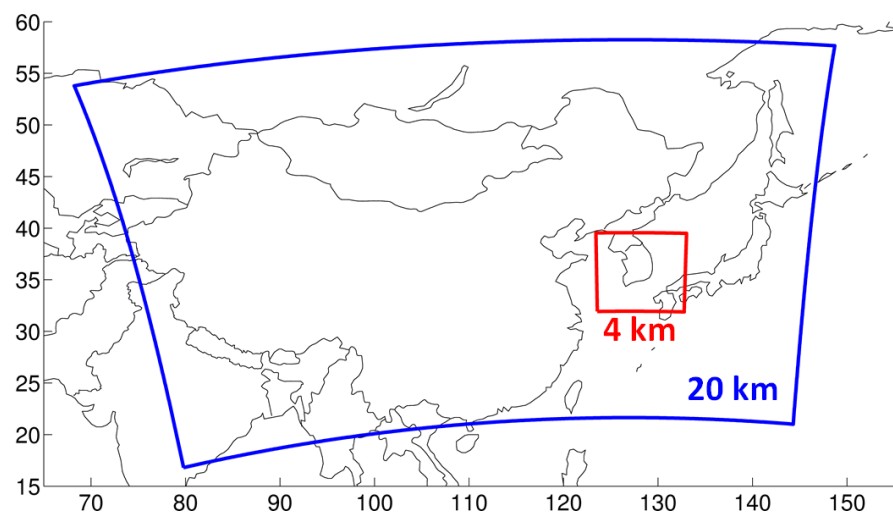

**Figure 1. Modeling domains for the forecast simulations. Only the outer domain is used for the retrospective simulations.**

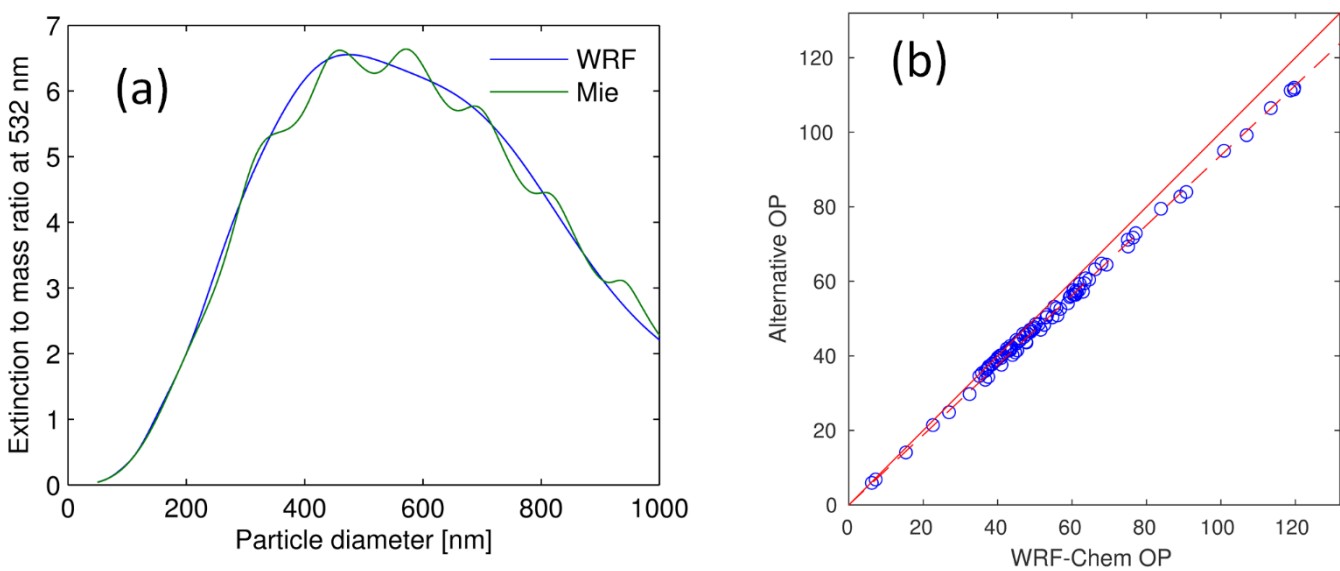

**Figure 2. a) Extinction to mass ratio for dry conditions (20% RH) as a function of dry particle diameter considering a**
980 **monodisperse aerosol distribution of fixed aerosol composition equal to the mean of the data analyzed. Blue and green lines represent results using optical properties code from WRF-Chem and the alternative approach, respectively. b) Scatter plot comparing aerosol water [µg/m³] estimated by WRF-Chem routines for the forecast simulations (MOSAIC 4 bin) and using the alternative approach at 80% RH during 2:00-5:00 UTC of the flight analyzed in this study. The solid red line indicates the 1:1 line and the dashed red line represents the regression line (slope of 0.94).**

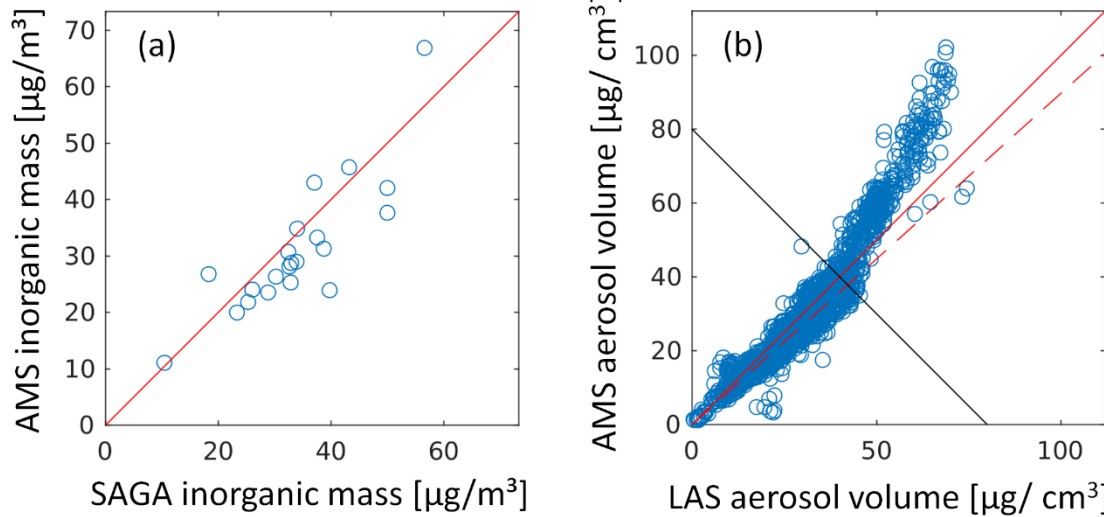

**Figure 3.** a) Scatterplot of total inorganic aerosol (Sulfate, ammonium, chloride and nitrate) mass [μg/m³] as measured by SAGA and AMS, averaging the AMS data to the SAGA integration time (R² = 0.72, slope = 1.07). The solid red line indicates the 1:1 line. b) Scatterplot of aerosol volume measured by the AMS and LAS volume accounting for the AMS transmission. The solid red line 990 indicates the 1:1 line, the solid black line represents an approximate cut-off for the LAS saturation, and the dashed red line represents the regression line when using data below the black line (slope of 0.9).

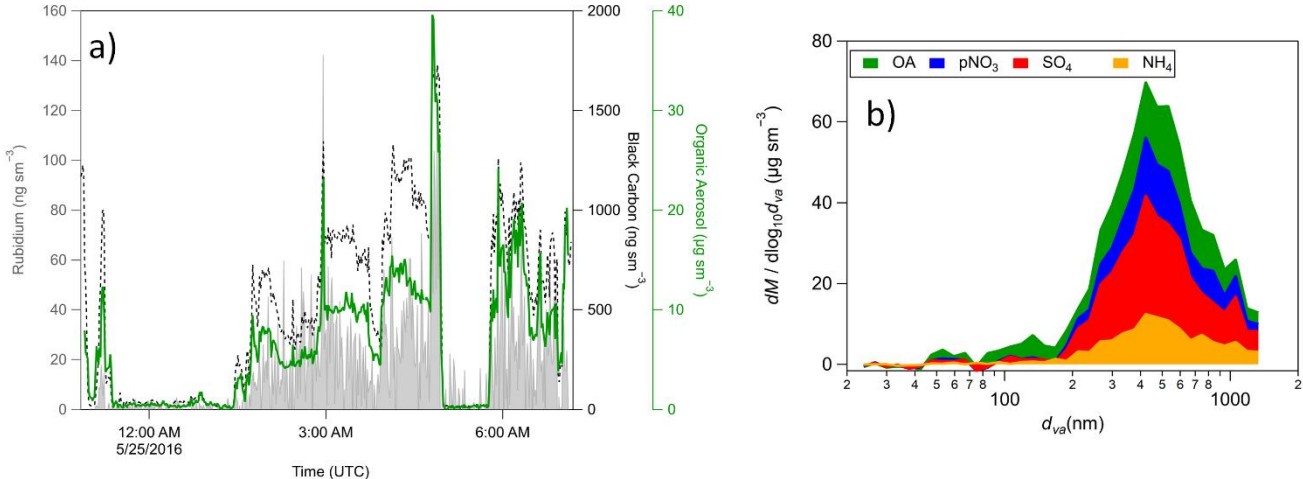

**Figure 4.** a) Time series of rubidium (grey, left axis, measured by the AMS), black carbon (black dashed line, right axis, measured by SP2), and total organic aerosol (green, right axis, measured by AMS), during the haze event sampled by the NASA DC-8 over 995 the Yellow Sea. Rubidium was quantified using the AMS difference signal, a relative ionization efficiency of 1, and the same collection efficiency as the rest of the submicron aerosol (Nault et al., 2018). b) Average size resolved AMS measurements sampled (aerodynamic diameter) by the NASA DC-8 for the same period shown in a).

31

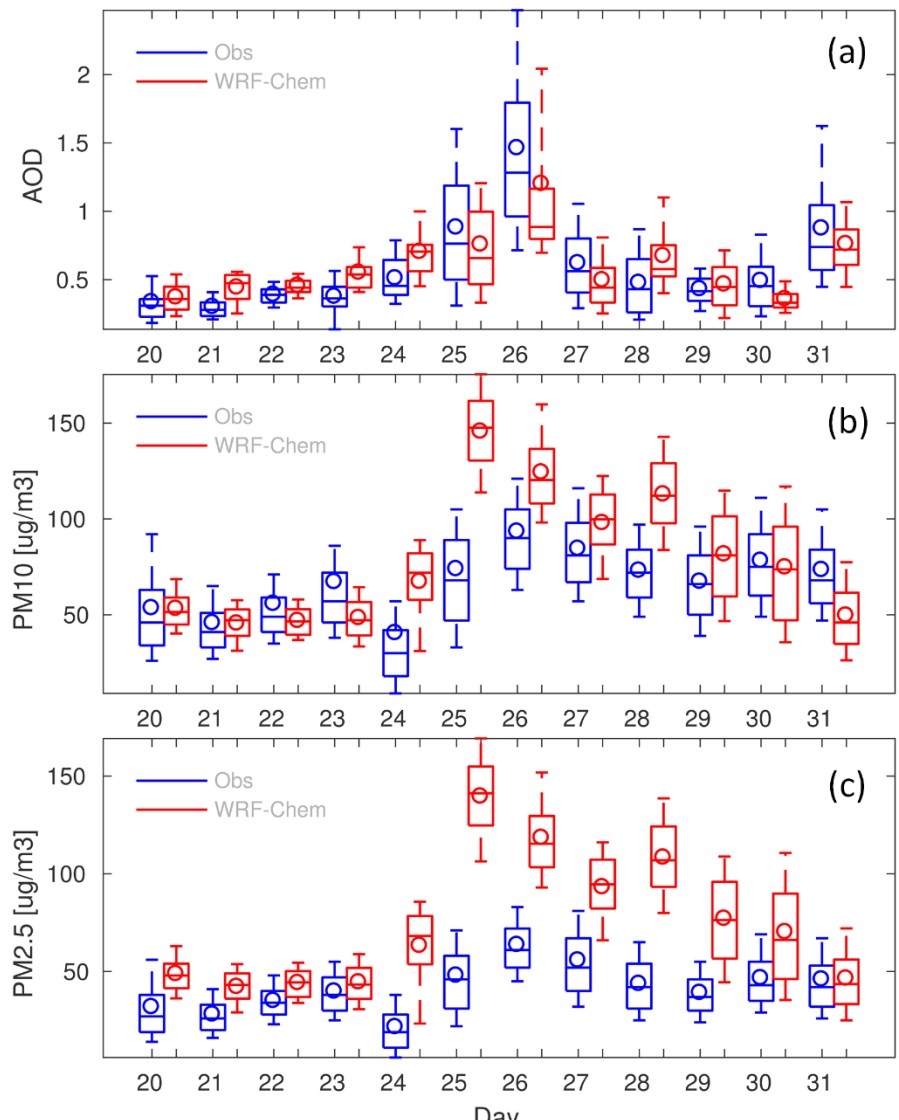

**Figure 5.** Time series of box and whisker plots for AOD (a), PM10 (b) and PM2.5 (c) for select days on the month of May 2016 comparing observations and forecasts over sites in South Korea. Data are aggregated by day (in UTC time). Center solid lines indicate the median, circles represent the mean, boxes indicate upper and lower quartiles, and whiskers show the upper and lower deciles.

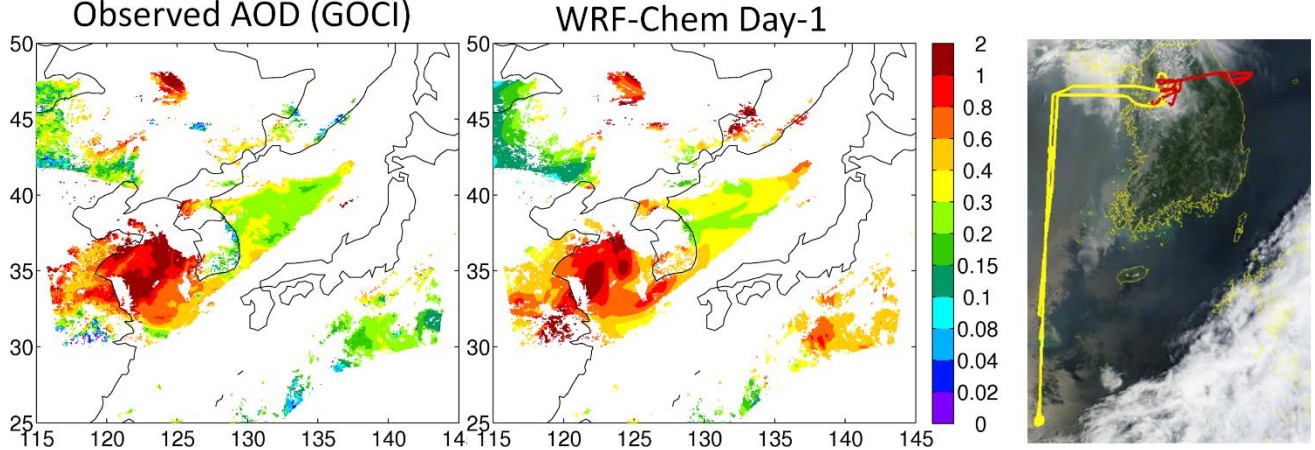

Figure 6. Observed (left) and forecasted (center) AOD maps at 3 UTC (noon local Korean time) on May 25th. The right plot shows Advanced Himawari Imager true color imagery for the same time with overlays of the DC-8 (in yellow) and Twin Otter (in red) flight tracks for that day (Source: KORUS-AQ flight report).

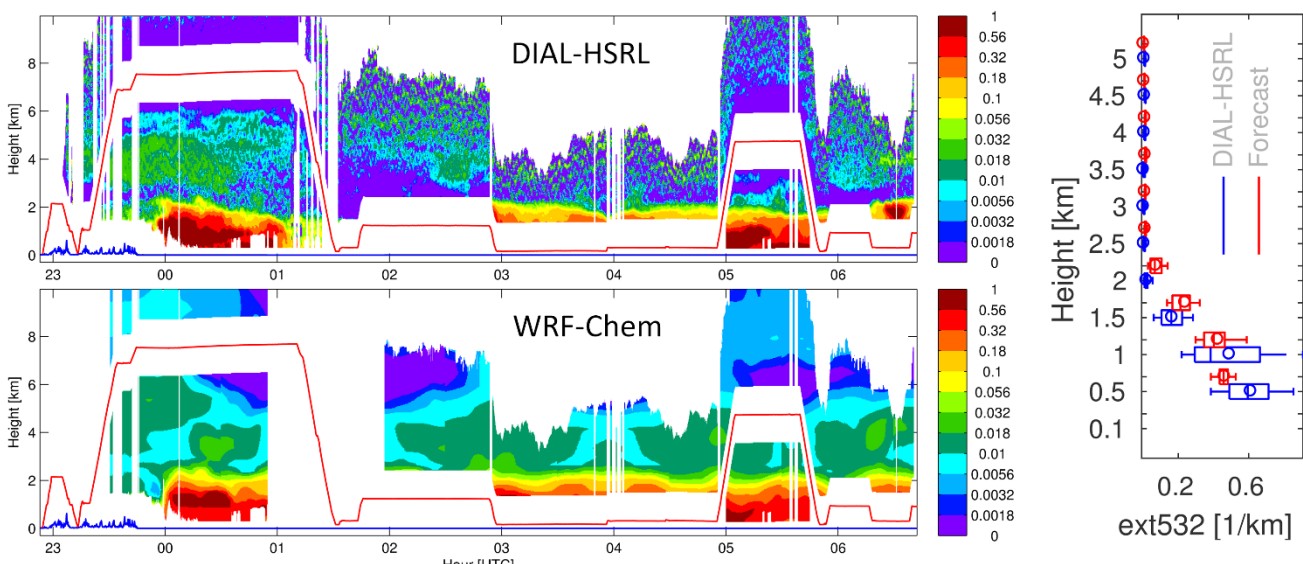

Figure 7. Left panels: DIAL-HSRL (upper panel) and WRF-Chem forecast simulation (bottom panel) extinction curtains at 532 nm for the KORUS-AQ flight on May 24th. The red solid line represents the altitude of the aircraft from where DIAL-HSRL was being operated. Right panel: Box and whisker plots as in Figure 5 aggregating observed and modeled data shown in the left panels for the periods where the aircraft was above the haze layer (00-01 UTC and 5-5:45 UTC).

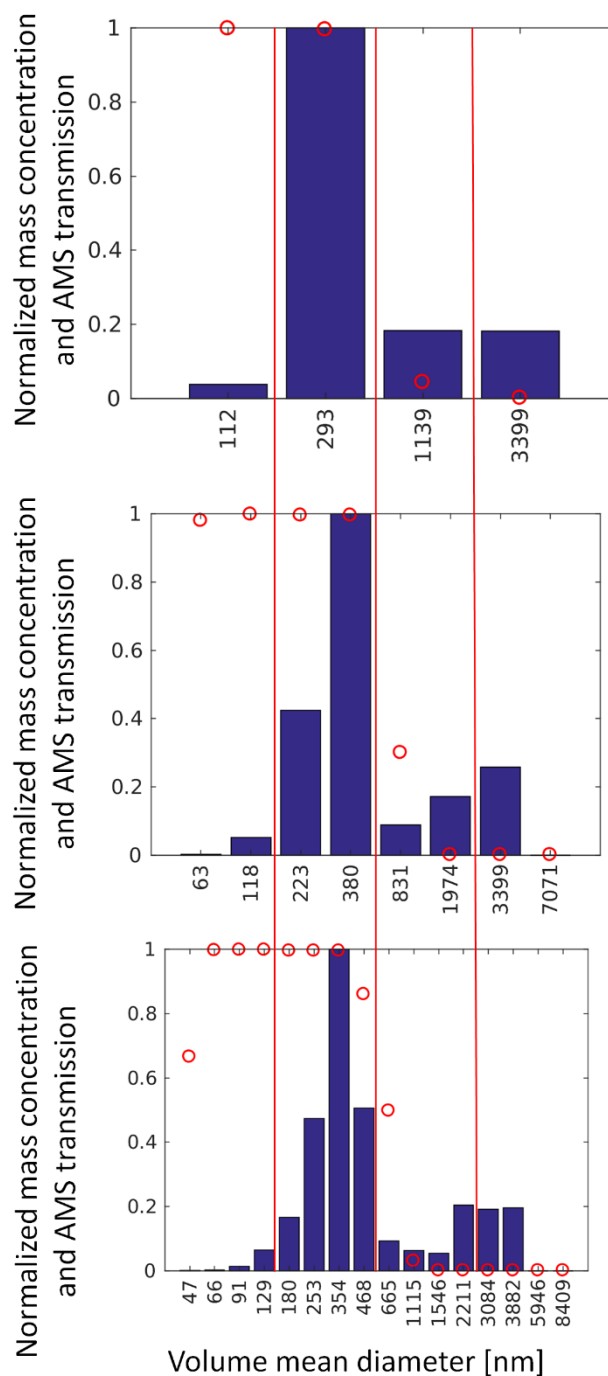

**Figure 8. Resulting average size distributions (blue bars) when aggregating observed data (2:00-5:00 UTC) to 4, 8 and 16 size bins. Sizes distributions show average mass concentration in each bin normalized by the maximum value within each distribution. Red lines separate bins aggregated when going from finer to coarser bin representation. Size bins boundaries are defined in Table 3. Red circles indicate the average AMS transmission efficiency (dimensionless) for each size bin.**

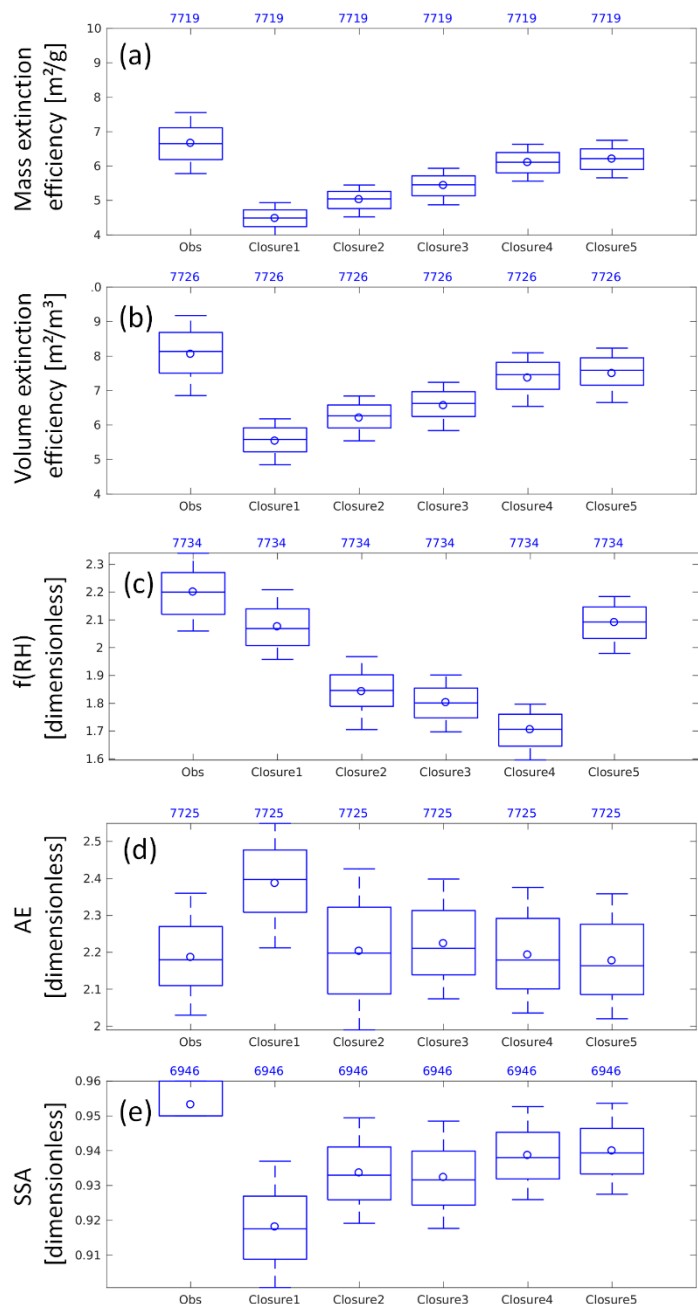

**Figure 9 Box and whisker plots (as in Fig. 5) showing observations and closure results driving the optical properties code with observations. Closure cases are described in Table 4. Results are shown for (a) the extinction (550 nm) to mass ratio (mass extinction efficiency), (b) extinction to volume ratio (volume extinction efficiency), (c) f(RH) measured at 550 nm, (d) 550-700 Ångström exponent and (e) Dry single-scattering albedo. The blue numbers on top of the plots represent the sample size used when computing statistics.**

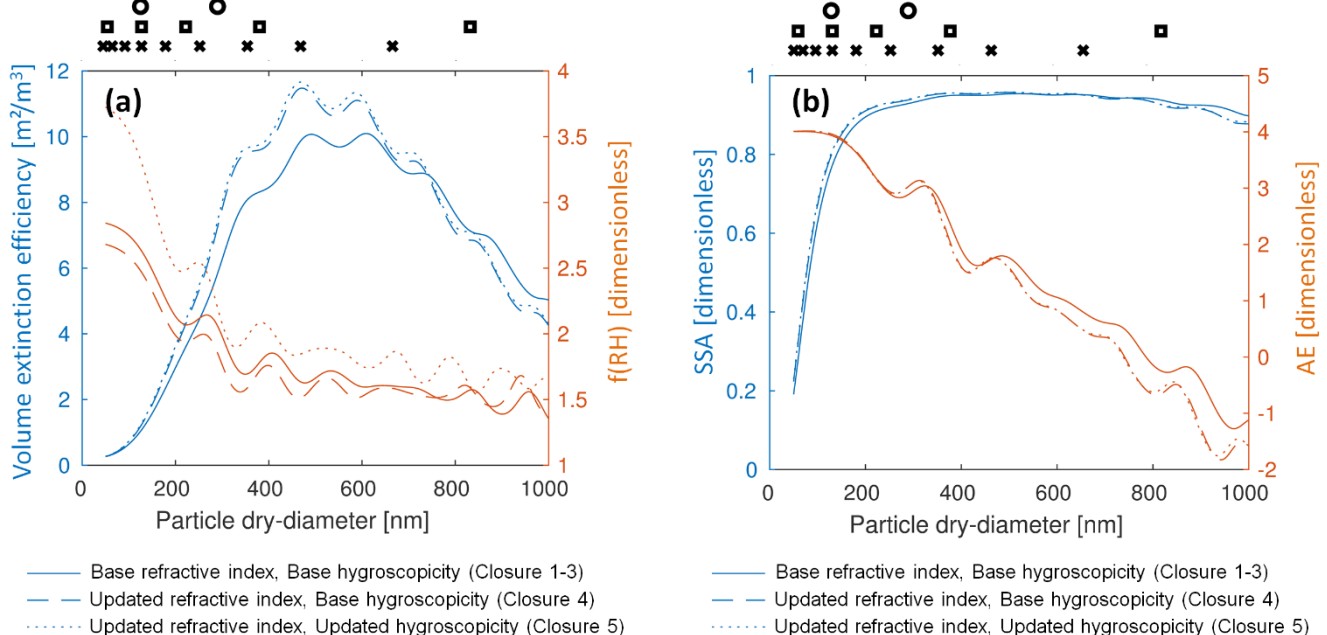

**Figure 10. a) Volume extinction efficiency (blue, scale on the left) and f(RH) (orange, scale on the right) as a function of geometric dry particle diameter considering a monodisperse aerosol distribution of fixed aerosol composition equal to the mean of the data analyzed. Different lines represent cases where real refractive index and hygroscopicity correspond to the base and updated conditions (see text for details). Black markers on top of the plots represent the calculated volume mean diameter for each size bin when using 4 (circles), 8 (squares) and 16 (x's) bins for the mean observed size distribution (numerical values found on Fig. 8).**

**Note only mean diameters below 1 μm are shown. b) Same as a) but for dry Single-scattering albedo and 550-700nm Ångström exponent.**

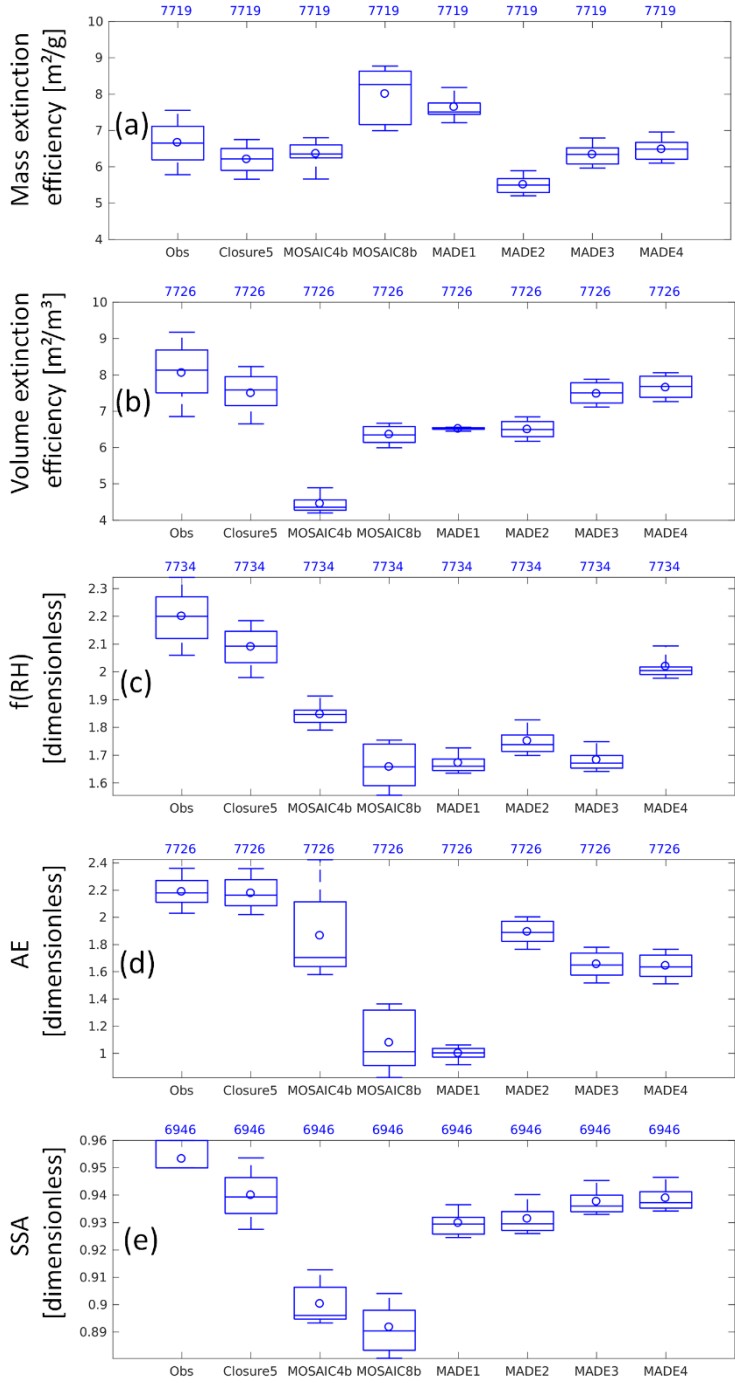

**Figure 11. As Figure 9 but comparing observations and Closure study 5 to different modeling configurations (described in Table 1).**

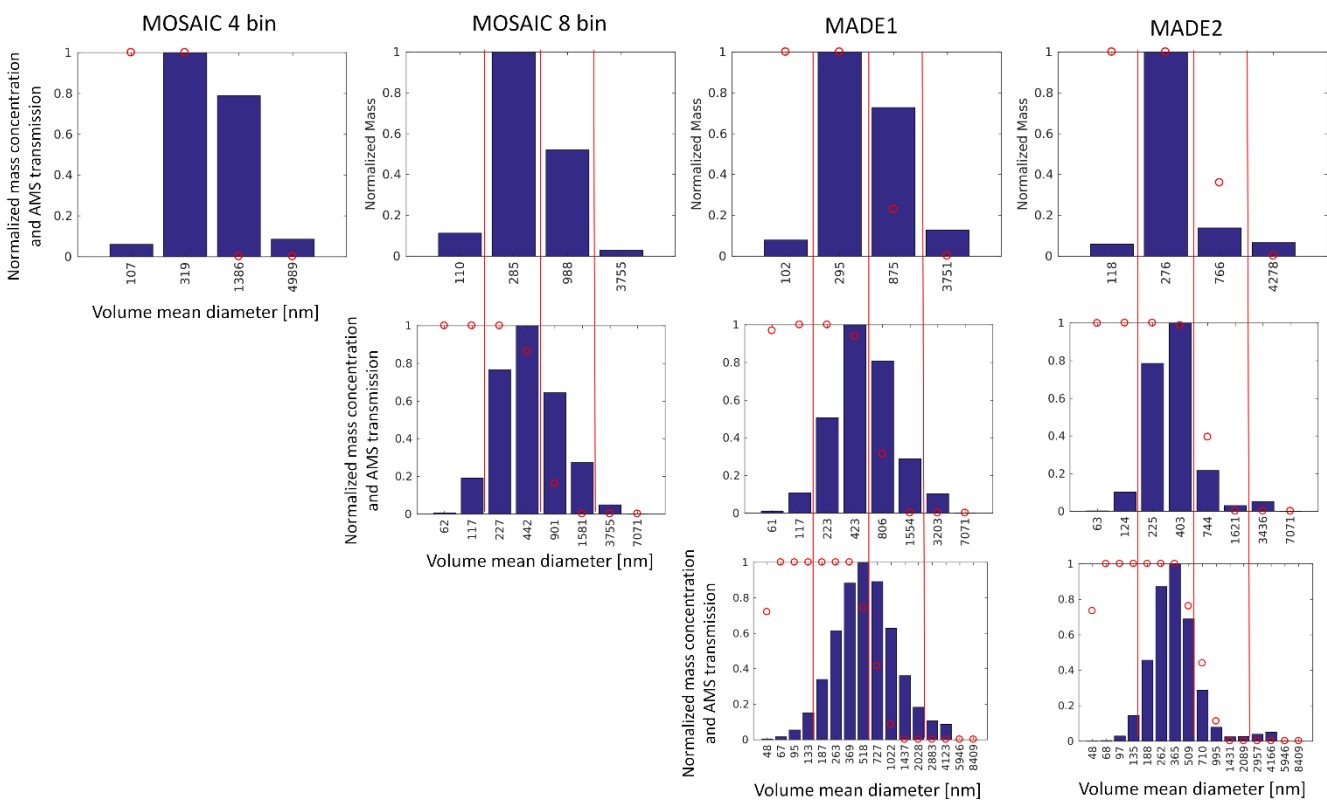

**Figure 12. Same as Figure 8 but for simulations using different modeling configurations. Size bins and AMS transmission efficiency are aggregated to coarser bins when possible for comparison across configurations. Size bin boundaries are defined in Table 3.**

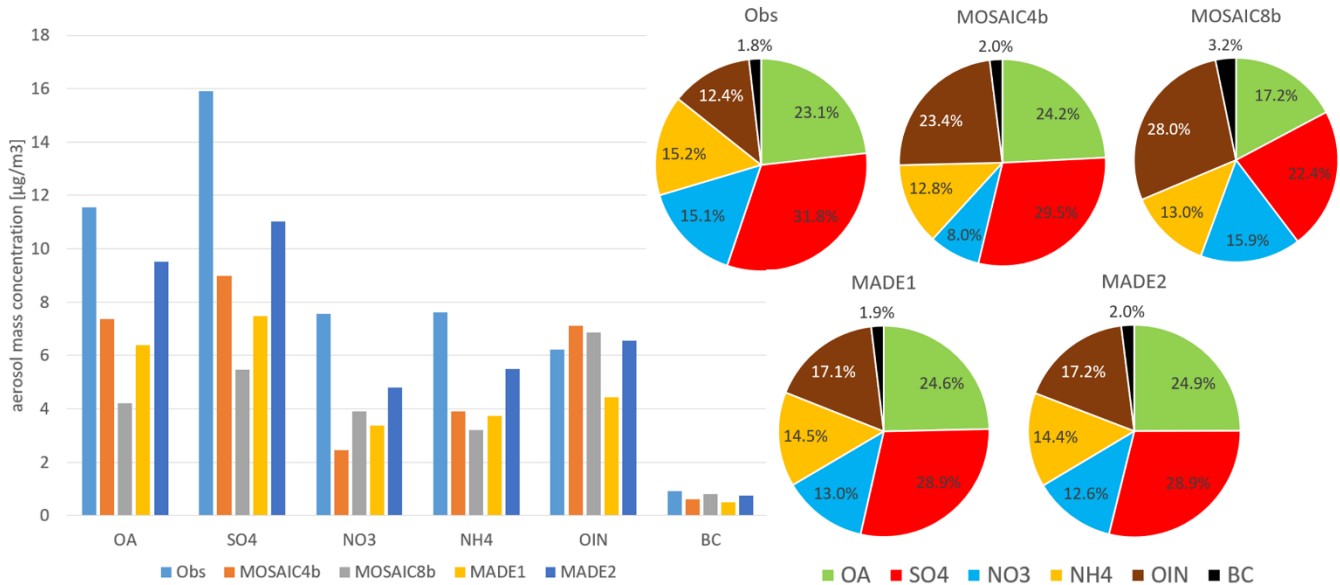

**Figure 13. Left panel: Bar plot showing sub-micron aerosol mass concentration by species for observations (AMS+SP2) and retrospective simulations. Right panels: Pie charts showing percentage contribution by each species for observations and retrospective simulations.**