# Peer review of "Understanding and improving model representation of aerosol optical properties for a Chinese haze event measured during KORUS-AQ"

_Atmospheric Chemistry and Physics, 2019_

## Referee Comment (RC1) · Anonymous Referee #1 · 4 Dec 2019

Review of article: "Understanding and improving model representation of aerosol optical properties for a Chinese haze event measured during KORUS-AQ" by Pablo E. Saide et al.

General comments

This work evaluates the deficiencies in the estimation of aerosol optical properties from aerosol mass in the WRF-Chem model. Within the international air quality field study KORUS-AQ, authors found out that aerosol optical depth (AOD) data assimilation works properly but surface particulate concentrations were over-predicted by WRF-Chem.

[Figure]

Following these results, the authors explain that these discrepancies can be due to 1) how well the model represents the aerosol properties which drive the optical properties computation (e.g., size distribution, composition, concentrations, etc.); and 2) the accuracy of the optical properties code.

First, authors scrutinize the accuracy of the optical properties by running this code using in-situ observations of size distribution and compositions as inputs. They found that a finer size bin representation and an update of refractive indices and hygroscopicity parameters make computed optical properties closer to the measured ones.

After that the authors tried to evaluate how the model represents the aerosol properties which drive the optical properties computations. With this objective, they run a set of simulations with different aerosol options and taking into account or not the results previously found. This experiment reveals the inability of sectional and modal aerosol configuration in WRF-Chem to properly reproduce the observed size distribution among the underestimation of organic aerosol density and the overprediction of the fractional contribution of inorganic aerosols other than those already taken into account.

Although, in my opinion this is an interesting work, I have found some important issues that deserve a major revision and could, in my opinion, improve the overall quality of this work.

Firstly, I think that a deeply revision of the Results and discussion section should be done in order to include numerical results. In my opinion, in this section authors correctly describes figures qualitatively and make a discussion of the results, but they do not provide a description of the numerical results found. Figures should be described indicating the numerical results found. For instant, in sections 3.1 and 3.2 authors described the results in Figure 9, but they should include the numerical results from observations and the different closure studies. In my opinion, this should be done with all figures and, in particular, through the Results and discussion section for a better

understanding of the work.

On the other hand, in section 2.1 the model setup is described but the authors do not indicate two important issues. How natural emissions, such as, desert dust or sea salt, or biogenic emissions are being considered by WRF-Chem? This should be mentioned and explained due to the high influence of this emissions on particulate matter.

Moreover, they do not indicate whether ARI and ACI were taken into account in the configuration of the simulations. Previous works, such as, Palacios-Peña et al., 2017; 2018 have demonstrated an improvement in the representation of AOD when these interactions were taken into account. This should be clarified and taken into account in the discussion of the results.

Finally, why authors use Level 1.5 of AERONET instead of Level 2.0 whose quality is assured? Authors may find useful to use data from the MAN network ( https://aeronet.gsfc.nasa.gov/new_web/maritime_aerosol_network.html ).

Specific Comments Abstract It would be useful to describe results join to some quantified results.

Introduction I recommend a revision and an improvement of this section. I attach in this review some references that could enhance this section.

• Lines 62-63: "Again, this translation of aerosol mass to optical properties is performed in these models, often showing large inter-model variability (Myhre et al., 2013; Stier et al., 2013)" Similar inter-model variability was found also by Kipling et a.,2016. • Lines 74-76: "Crippa et al. (2019) performed an ensemble of simulations to assess what combination of model inputs and configurations resulted in the best agreement to observations in the southeast US." Palacios-Peña et al. 2019 and Curci et al. 2019 also investigated aerosol optical properties representation over Europe. Methods • 2.1. Regional modelling: Lines 130-135: ". . . WRF-Chem can also be configured with the Modal Aerosol Dynamics Model for Europe (MADE) model, where aerosol sizes are

represented by log-normal modes (as opposed to sections as for MOSAIC). We used the configuration coupled to the updated Regional Atmospheric Chemistry Mechanism (RACM, Ahmadov et al., 2014) which contains secondary organic aerosol formation using the volatility basis-set (Ahmadov et al., 2012) and aerosol optical properties calculations (Tuccella et al., 2015). We label these simulations as RACM#, with # going from 1-4 depending on changes to parameters described in Table 1." For an easy understanding of the manuscript, I would recommend a rename of the label for the modal distribution. Instead of use RACM#, I would use MADE#. This is because the aerosol module is MADE and not RACM which is the gas-phase module. This could lead to a mis-understating in reading.   2.1. Regional modelling: Lines 137-138: "All retrospective simulations were performed only for the 20 km resolution domain in this study, as we focus on a pollution event from long range transport." Please remind the date of the episode in this part of the text.   2.2. Optical properties calculation. Line 150 and somewhere hereinafter. The authors indicate that off-line versions of optical properties calculation from WRF-Chem are needed. What do you mean with off-line versions? Do you mean without ARI and ACI? This should be clarified.   2.3. Airborne Observations. Line 175: What is the meaning of NASA DC-8? Please clarify. The authors should be careful with this kind of nomenclature, in particular, taking into account non specialist observational readers.

Results   Lines 432-434: "Another point to note is that models under-predict the relative magnitude of the coarse aerosols (2.5-10 $\mu$m range, bin #4 in the 4-bin configuration). This helps to explain why the biases shown in Figure 5 are more pronounced for PM2.5 than PM10, as the under-prediction in the coarse aerosols is offset by the over-prediction in the fine aerosols." Similar results were found by Balzarini et al. 2015 and Im et al. 2015.   Figure 7: What represents the red line? This should be clarified both in text and in figure caption.

Technical Comments (of purely technical corrections at the very end: typing errors, etc.)

[Figure]

• Through the text: "Angstrom" should be corrected by "Ånsgtröm". • Check for the parenthesis related with acronyms and cites. Some examples are: o Line 107: "geostationary satellites (Geostationary Ocean Color Imager retrievals (Choi et al., 2018; Choi et al., 2016))" should be changed by "geostationary satellites (Geostationary Ocean Color Imager retrievals; Choi et al., 2018; Choi et al., 2016)". o Line 183: "High-Resolution Time-of-Flight Aerosol Mass Spectrometer (HR-ToF-AMS, hereinafter "AMS" for short) (DeCarlo et al., 2006; Nault et al., 2018)." should be changed by "High-Resolution Time-of-Flight Aerosol Mass Spectrometer (HR-ToF-AMS, hereinafter "AMS" for short; DeCarlo et al., 2006; Nault et al., 2018)." o Line 186: "Particle Soot Photometer (SP2) (Lamb et al., 2018)." should be changed by "Particle Soot Photometer (SP2; Lamb et al., 2018)." o Line 192: "Particle Soot Absorption Photometer (at 470, 532, and 660nm wavelength) (Ziemba et al., 2013)." should be changed by "Particle Soot Absorption Photometer (at 470, 532, and 660nm wavelength, Ziemba et al., 2013)." o Line 198: "High Spectral Resolution Lidar (HSRL) (Hair et al., 2008)" should be changed by "High Spectral Resolution Lidar (HSRL; Hair et al., 2008)". o Line 228: "Rubidium originates either from soil (e.g., dust) (Kabata-Pendias and Pendias, 2001)" should be changed by "Rubidium originates either from soil (e.g., dust; Kabata-Pendias and Pendias, 2001)" o Line 297: "KORUS-AQ flights (and thus not detected by AMS) (Heim et al., 2019)," should be changed by "KORUS-AQ flights (and thus not detected by AMS; Heim et al., 2019)," • Line 186: "of New Hampshire usingTeflon filters" add a gap "of New Hampshire using Teflon filters".

Bibliography Balzarini, A., Pirovano, G., Honzak, L., Žabkar, R., Curci, G., Forkel, R., ... & Grell, G. A. (2015). WRF-Chem model sensitivity to chemical mechanisms choice in reconstructing aerosol optical properties. Atmospheric Environment, 115, 604-619. Curci, G., Alyuz, U., Barò, R., Bianconi, R., Bieser, J., Christensen, J. H., Colette, A., Farrow, A., Francis, X., Jiménez-Guerrero, P., Im, U., Liu, P., Manders, A., Palacios-Peña, L., Prank, M., Pozzoli, L., Sokhi, R., Solazzo, E., Tuccella, P., Unal, A., Vivanco, M. G., Hogrefe, C., and Galmarini, S.: Modelling black carbon absorption of solar radiation: combining external and internal mixing assumptions,

Atmos. Chem. Phys., 19, 181–204, https://doi.org/10.5194/acp-19-181-2019, 2019. Im, U., Bianconi, R., Solazzo, E., Kioutsioukis, I., Badia, A., Balzarini, A., ... & Curci, G. (2015). Evaluation of operational on-line-coupled regional air quality models over Europe and North America in the context of AQMEII phase 2. Part I: Ozone. Atmospheric Environment, 115, 404-420. Kipling, Z., Stier, P., Johnson, C. E., Mann, G. W., Bellouin, N., Bauer, S. E., Bergman, T., Chin, M., Diehl, T., Ghan, S. J., Iversen, T., Kirkevåg, A., Kokkola, H., Liu, X., Luo, G., van Noije, T., Pringle, K. J., von Salzen, K., Schulz, M., Seland, Ø., Skeie, R. B., Takemura, T., Tsigaridis, K., and Zhang, K.: What controls the vertical distribution of aerosol? Relationships between process sensitivity in HadGEM3–UKCA and inter-model variation from AeroCom Phase II, Atmos. Chem. Phys., 16, 2221–2241, https://doi.org/10.5194/acp-16-2221-2016, 2016. Palacios-Peña, L., Baró, R., Guerrero-Rascado, J. L., Alados-Arboledas, L., Brunner, D., and Jiménez-Guerrero, P.: Evaluating the representation of aerosol optical properties using an online coupled model over the Iberian Peninsula, Atmos. Chem. Phys., 17, 277–296, https://doi.org/10.5194/acp-17-277-2017, 2017. Palacios-Peña, L., Baró, R., Baklanov, A., Balzarini, A., Brunner, D., Forkel, R., Hirtl, M., Honzak, L., López-Romero, J. M., Montávez, J. P., Pérez, J. L., Pirovano, G., San José, R., Schröder, W., Werhahn, J., Wolke, R., Žabkar, R., and Jiménez-Guerrero, P.: An assessment of aerosol optical properties from remote-sensing observations and regional chemistry–climate coupled models over Europe, Atmos. Chem. Phys., 18, 5021–5043, https://doi.org/10.5194/acp-18-5021-2018, 2018. Palacios-Peña, L., Jiménez-Guerrero, P., Baró, R., Balzarini, A., Bianconi, R., Curci, G., Landi, T. C., Pirovano, G., Prank, M., Riccio, A., Tuccella, P., and Galmarini, S.: Aerosol optical properties over Europe: an evaluation of the AQMEII Phase 3 simulations against satellite observations, Atmos. Chem. Phys., 19, 2965–2990, https://doi.org/10.5194/acp-19-2965-2019, 2019.

Please also note the supplement to this comment:
https://www.atmos-chem-phys-discuss.net/acp-2019-1022/acp-2019-1022-RC1-

supplement.pdf

---

## Referee Comment (RC2) · Anonymous Referee #2 · 12 Dec 2019

This manuscript provides a detailed analysis of how well a model simulation of a pollution event in the vicinity of the Korean peninsula compares with detailed in situ and remote sensing measurements. The authors do a good job of making use of available airborne, surface, and remote-sensing data sets and multiple model configurations to carefully consider why there was a mismatch between modeled and measured aerosol optical depth (which agreed well) and aerosol mass concentrations (which disagreed by a factor of $\sim$2). This is an important issue; many of the same parameterizations and assumptions found in the high-resolution WRF-Chem model are also used in global chemistry-climate models that estimate aerosol-cloud and aerosol-radiation interactions. Thus careful analysis of detailed case such as this can result in improvements

in model performance for climate studies. And improved model performance to better predict aerosol-health effects is also an extremely important topic. Thus this paper is of interest to a broad spectrum of ACP readers and is entirely appropriate for publication there.

The paper is quite thorough, well written, and clear. I especially appreciate the effort the authors have gone to appropriately compare the model output with the measurements, for example by applying the AMS sampling efficiency curve to the modeled size distributions. The sensitivity of the results to model bin width and to assumptions about hygroscopicity and modal width are important and are especially well described and highlighted. There are few small clarifications needed, as described below. After these minor edits, the paper should be ready for final publication.

Minor Comments:

1) Line 102: Extra space at the end of the sentence. Please run a spell checker to find other small typographic errors that persist. Also please look for occasional random capitalizations of nouns. 2) Line 175: Explain what the "NASA DC-8" is. 3) Fig. 4. The grey trace showing rubidium obscures the underlying BC and OA traces. Can you lighten this or show it as dots rather than as a shaded region? Also please make one of the other traces a dashed line to accommodate color-blind readers. Please check other figures for the same issues. 4) Fig. 9e, the left-most box-and-whisker plot showing measured SSA is off-scale. 5) Fig. 10a, the axis label says "mass extinction efficiency" but the caption says "volume extinction efficiency". I also recommend you plot these parameters on a log scale on the x-axis, as for the other graphs showing size-dependent aerosol properties such as size distribution. 6) On Figs. 8 and 12, you may want to divide the bar height by the logarithmic bin width to put these size distributions on a dN/dlogDp scale. You then don't have to change the y-axis scale and the reader can see that the re-binning conserves the size distribution number.

---

## Author Comment (AC1) · 13 Apr 2020

**Here find explanation of the changes made to the manuscript from the previous version. These were**
**made both in response to the two referees, and due to additional realizations and input which is**
**described at the end of this document.**

**Response to referee #1**

**General comments**

**Referee #1:** This work evaluates the deficiencies in the estimation of aerosol optical properties from
aerosol mass in the WRF-Chem model. Within the international air quality field study KORUS-AQ,
authors found out that aerosol optical depth (AOD) data assimilation works properly but surface
particulate concentrations were over-predicted by WRFChem.

Following these results, the authors explain that these discrepancies can be due to 1) how well the
model represents the aerosol properties which drive the optical properties computation (e.g., size
distribution, composition, concentrations, etc.); and 2) the accuracy of the optical properties code.

First, authors scrutinize the accuracy of the optical properties by running this code using in-situ
observations of size distribution and compositions as inputs. They found that a finer size bin
representation and an update of refractive indices and hygroscopicity parameters make computed
optical properties closer to the measured ones.

After that the authors tried to evaluate how the model represents the aerosol properties which drive
the optical properties computations. With this objective, they run a set of simulations with different
aerosol options and taking into account or not the results previously found. This experiment reveals the
inability of sectional and modal aerosol configuration in WRF-Chem to properly reproduce the observed
size distribution among the underestimation of organic aerosol density and the overprediction of the
fractional contribution of inorganic aerosols other than those already taken into account.

Although, in my opinion this is an interesting work, I have found some important issues that deserve a
major revision and could, in my opinion, improve the overall quality of this work.

**Authors:** We really appreciate your review, we tried to incorporate your suggestions as outlined below.

**Referee #1:** Firstly, I think that a deeply revision of the Results and discussion section should be done in
order to include numerical results. In my opinion, in this section authors correctly describes figures
qualitatively and make a discussion of the results, but they do not provide a description of the numerical
results found. Figures should be described indicating the numerical results found. For instant, in sections
3.1 and 3.2 authors described the results in Figure 9, but they should include the numerical results from
observations and the different closure studies. In my opinion, this should be done with all figures and, in
particular, through the Results and discussion section for a better understanding of the work.

**Authors:** The results and discussion section has been modified to include more numerical results in all
its subsections.

**Referee #1:** On the other hand, in section 2.1 the model setup is described but the authors do not
indicate two important issues. How natural emissions, such as, desert dust or sea salt, or biogenic
emissions are being considered by WRF-Chem? This should be mentioned and explained due to the high
influence of this emissions on particulate matter.

**Authors:** All of these emissions treatments including that of biomass burning are now included in
section 2.1.

**Referee #1:** Moreover, they do not indicate whether ARI and ACI were taken into account in the
configuration of the simulations. Previous works, such as, Palacios-Peña et al., 2017; 2018 have
demonstrated an improvement in the representation of AOD when these interactions were taken into
account. This should be clarified and taken into account in the discussion of the results.

**Authors:** This is now included in section 2.1: "Aerosol-radiation interactions were included (Fast et al.,
2006), while aerosol-cloud interactions were excluded to avoid the computational costs of tracking the
cloud-borne aerosols.".  Since all simulations were performed with the same ARI/ACI setups, ARI/ACI
were not a focus of the sensitivity simulations, and do not affect our findings.

**Referee #1:** Finally, why authors use Level 1.5 of AERONET instead of Level 2.0 whose quality is assured?
Authors may find useful to use data from the MAN network
(https://aeronet.gsfc.nasa.gov/new_web/maritime_aerosol_network.html ).

**Authors:** At the time we downloaded the data there were still some sites where there was no level 2
data yet so level 1.5 was used. We checked now and all data is processed at level 2 so the AOD plot in
Figure 5 was updated. The data are nearly unchanged so there are no changes to the manuscript.

Regarding the MAN network, there were two vessels that were present during the KORUS-AQ period in
the area (RV Jangmok and Onnuri) but they were towards the south of the Korean peninsula, so they did
not capture the highest aerosol loads of the pollution event studied here (see Figure 6).

**Specific Comments**

**Referee #1:** Abstract: It would be useful to describe results join to some quantified results.

**Authors:** Quantitative results were added to the abstract.

**Introduction**

**Referee #1:** I recommend a revision and an improvement of this section. I attach in this review some references that could enhance this section.

• Lines 62-63: "Again, this translation of aerosol mass to optical properties is performed in these models, often showing large inter-model variability (Myhre et al., 2013; Stier et al., 2013)" Similar inter-model variability was found also by Kipling et a.,2016.

• Lines 74-76: "Crippa et al. (2019) performed an ensemble of simulations to assess what combination of model inputs and configurations resulted in the best agreement to observations in the southeast US." Palacios-Peña et al. 2019 and Curci et al. 2019 also investigated aerosol optical properties representation over Europe.

**Authors:** We apologize for this oversight, description of these references is now included in the Introduction.

**Methods**

**Referee #1:** 2.1. Regional modelling: Lines 130-135: "… WRF-Chem can also be configured with the Modal Aerosol Dynamics Model for Europe (MADE) model, where aerosol sizes are represented by log-normal modes (as opposed to sections as for MOSAIC). We used the configuration coupled to the updated Regional Atmospheric Chemistry Mechanism (RACM, Ahmadov et al., 2014) which contains secondary organic aerosol formation using the volatility basis-set (Ahmadov et al., 2012) and aerosol optical properties calculations (Tuccella et al., 2015). We label these simulations as RACM#, with # going from 1-4 depending on changes to parameters described in Table 1." For an easy understanding of the manuscript, I would recommend a rename of the label for the modal distribution. Instead of use RACM#, I would use MADE#. This is because the aerosol module is MADE and not RACM which is the gas-phase module. This could lead to a mis-understating in reading.

**Authors:** Good point, we made this change throughout the text and figures.

**Referee #1:** 2.1. Regional modelling: Lines 137-138: "All retrospective simulations were performed only for the 20 km resolution domain in this study, as we focus on a pollution event from long range transport." Please remind the date of the episode in this part of the text.

**Authors:** Dates were added

**Referee #1:** 2.2. Optical properties calculation. Line 150 and somewhere hereinafter. The authors
indicate that off-line versions of optical properties calculation from WRF-Chem are needed. What do
you mean with off-line versions? Do you mean without ARI and ACI? This should be clarified.

**Authors:** This is indeed confusing, so we modified to "computations at the post-processing stage"
throughout the text. The question about ARI and ACI configuration was addressed via a previous
response to Reviewer #1.

**Referee #1:** 2.3. Airborne Observations. Line 175: What is the meaning of NASA DC-8? Please clarify. The
authors should be careful with this kind of nomenclature, in particular, taking into account non specialist
observational readers.

**Authors:** It's the name of the aircraft. This was modified to: "Airborne data used in this study were
measured by instruments on board of the NASA DC-8 research aircraft as part of the KORUS-AQ
campaign"

**Results**

**Referee #1:** Lines 432-434: "Another point to note is that models under-predict the relative magnitude
of the coarse aerosols (2.5-10 μm range, bin #4 in the 4-bin configuration). This helps to explain why the
biases shown in Figure 5 are more pronounced for PM2.5 than PM10, as the under-prediction in the
coarse aerosols is offset by the over-prediction in the fine aerosols." Similar results were found by
Balzarini et al. 2015 and Im et al. 2015.

**Authors:** These references were added supporting this statement.

**Referee #1:** Figure 7: What represents the red line? This should be clarified both in text and in figure
caption.

**Authors:** The red line indicates the altitude of the aircraft. This was added in the caption and in the text.

**Technical Comments (of purely technical corrections at the very end: typing errors, etc.)**

**Referee #1:** Through the text: "Angstrom" should be corrected by "Ånsgtröm".

**Authors:** This was corrected throughout the text

**Referee #1:** Check for the parenthesis related with acronyms and cites (see reviewer doc for full text).

**Authors:** All of these were changed

**Referee #1:** Line 186: "of New Hampshire usingTeflon filters" add a gap "of New Hampshire using Teflon
filters".

**Authors:** Corrected

**Response to Referee #2**

**Referee #2:** This manuscript provides a detailed analysis of how well a model simulation of a pollution event in the vicinity of the Korean peninsula compares with detailed in situ and remote sensing measurements. The authors do a good job of making use of available airborne, surface, and remote-sensing data sets and multiple model configurations to carefully consider why there was a mismatch between modeled and measured aerosol optical depth (which agreed well) and aerosol mass concentrations (which disagreed by a factor of ~2). This is an important issue; many of the same parameterizations and assumptions found in the high-resolution WRF-Chem model are also used in global chemistry-climate models that estimate aerosol-cloud and aerosol-radiation interactions. Thus careful analysis of detailed case such as this can result in improvements in model performance for climate studies. And improved model performance to better predict aerosol-health effects is also an extremely important topic. Thus this paper is of interest to a broad spectrum of ACP readers and is entirely appropriate for publication there.

The paper is quite thorough, well written, and clear. I especially appreciate the effort the authors have gone to appropriately compare the model output with the measurements, for example by applying the AMS sampling efficiency curve to the modeled size distributions. The sensitivity of the results to model bin width and to assumptions about hygroscopicity and modal width are important and are especially well described and highlighted. There are few small clarifications needed, as described below. After these minor edits, the paper should be ready for final publication.

**Authors:** We appreciate your comments, see our specific responses below.

**Referee #2:** 1) Line 102: Extra space at the end of the sentence. Please run a spell checker to find other small typographic errors that persist. Also please look for occasional random capitalizations of nouns.

**Authors:** We ran the spell checker and corrected typos throughout the manuscript.

**Referee #2:** 2) Line 175: Explain what the "NASA DC-8" is.

**Authors:** Modified to: "Airborne data used in this study were measured by instruments on board of the NASA DC-8 research aircraft as part of the KORUS-AQ campaign (Aknan and Chen, 2019) during the flight starting at 22:00 UTC on May 24$^{th}$ (May 25$^{th}$ in local Korean time) 2016."

**Referee #2:** 3) Fig. 4. The grey trace showing rubidium obscures the underlying BC and OA traces. Can you lighten this or show it as dots rather than as a shaded region? Also please make one of the other traces a dashed line to accommodate color-blind readers. Please check other figures for the same issues.

**Authors:** This figure has been updated according to the reviewer suggestion. Other figures with red and
blue should be readable by colorblind people. In Fig. 13, since bars and pie charts are in order, they
should also be readable by colorblind people.

**Referee #2:** 4) Fig. 9e, the left-most box-and-whisker plot showing measured SSA is off-scale.

**Authors:** The observed SSA values were stored with 2 significant digits. Looking back at the data there is
less than 3% with values over 0.96 and less than 10% below 0.95. This results in the whiskers and the
boxes having the same value, so the whiskers don't show up when this happens. Since the upper value
of the whisker and boxes is 0.96 then the plot it's not out scale. We prefer to keep the plot as is rather
than expand the scale to maximize the usage of space in the plot.

**Referee #2:** 5) Fig. 10a, the axis label says "mass extinction efficiency" but the caption says "volume
extinction efficiency". I also recommend you plot these parameters on a log scale on the x-axis, as for
the other graphs showing size-dependent aerosol properties such as size distribution.

**Authors:** Good catch! It was "volume extinction efficiency", there was a problem with the label but the
plot shows the right values and the discussion is based on the right variable. With respect to the x-axis
scale, we decided for the linear scale to better show the detail around the peak volume extinction
efficiency which is of importance for this study, it would be harder to read in the log scale. We also kept
it <1 um for the same reason, 0-1 um region would be too narrow if we wanted to cover up to 5 um
sizes. This is also the way this type of plots have been displayed in previous papers (see Fig A1 in Brock
et al., 2016a).

**Referee #2:** 6) On Figs. 8 and 12, you may want to divide the bar height by the logarithmic bin width to
put these size distributions on a dN/dlogDp scale. You then don't have to change the y-axis scale and the
reader can see that the re-binning conserves the size distribution number.

**Authors:** We tried Fig 8 using dMass/dlnD (see below) and the y-axis scale is still not consistent. Since
total mass concentrations are already in another figure, we decided to normalize the size distributions
to the maximum in each plot so all scales are the same (both for observations and model). Additionally,
we added AMS transmission efficiency to the modeled size distributions.

[Figure]

**Changes made outside of the scope of direct response to reviewers:**

We realized there was a problem in the time axis in Figure 5 for PM10 and PM2.5 (the first time interval
was 12 hours instead of a day). This is now corrected and time is consistent between all plots. This
change does not impact the results of the manuscript.

We also got the following comment on our discussion paper from a non-referee colleague through
email:

*Your Table 2 with the revised parameters got my special interest. I wanted to bring up that the value for*
*sodium chloride is at RH=90% not 1.16 but rather around 1.5. It got erroneously implemented into the*
*Petters and Kreidenweis paper from 2007 (see our 2017 paper on sea salt hygroscopicity:*
*https://www.nature.com/articles/ncomms15883 , page 5, right column and Table 1).*

*Interestingly, the value is quite close to the kappa-value of inorganic sea salt, which is the component*
*you would expect in ambient atmosphere. So the models are using the right value due to the wrong*
*reason when it comes to sea salt (ECHAM e.g. has the same issue). We discussed this bug in our 2017*
*paper and you might want to have a look at it. It would help if you would mention this aspect in your*
*table, otherwise this wrong value for NaCl gets further used. It would generally help if you give*
*references to the other values in your table (where possible).*

With this comment we realized that, while we were using the 1.16 hygroscopicity in our base alternative
approach, WRF-Chem routines associated to the MOSAIC model that compute aerosol water use the
correct properties of sodium chloride. Thus, for the base approach we changed sodium chloride
hygroscopicity parameter to 1.5 to make it consistent with MOSAIC and this accounted for a fraction of
the discrepancy found in Figure 2b. We updated this plot and the text and now the discrepancy is
reduced from ~10% to ~7%. For the updated values of hygroscopicity we used the value proposed by
Ziegler et al (2017) of 1.1. Since sea-salt was a minor component of aerosol composition (<1% in
observation and models) these changes did not have an impact on the results of this study.